

# Using Low Cost Sensors to Measure Ambient Particulate Matter Concentrations and On-Road Emissions Factors

Karoline K. Johnson[1], Michael H. Bergin[1], Armistead G. Russell[2], Gayle S. W. Hagler[3]

[1]School of Civil and Environmental Engineering, Duke University, Durham, NC, 27708, USA
[2] School of Civil and Environmental Engineering, Georgia Institute of Technology, Atlanta, GA, 30332, USA
[3] U.S. Environmental Protection Agency, Office of Research and Development, Research Triangle Park, NC, 27711, USA

*Correspondence to*: Karoline K. Johnson (Karoline.johnson@duke.edu)

**Abstract.** Air quality is a growing public concern in both developed and developing countries, as is the public interest in having information on air pollutant concentrations within their communities. Quantifying the spatial and temporal variability of ambient fine particulate matter ($PM_{2.5}$) is of particular importance due to the well-defined health impacts associated with $PM_{2.5}$. This work evaluates a number of select PM sensors (Shinyei: models PPD42NS, PPD20V, PPD60PV) under a variety of ambient conditions and locations including urban background and roadside sites in Atlanta, GA, as well as a location with substantially higher ambient concentrations in Hyderabad, India. Low cost sensor measurements were compared against reference monitors at all locations. On-road emissions factors were calculated at the Atlanta site by pairing $PM_{2.5}$ and separately determined black carbon (BC) and carbon dioxide ($CO_2$) measurements. On-road emission factors can vary in different locations and over time for a number of reasons, including vehicle fleet composition and driving patterns and behaviors, and current environmental policy. Emission factors can provide valuable information to inform researchers, citizens, and policy makers. The PPD20V sensors had the highest correlation with the reference environmental beta attenuation monitor (E-BAM) with $R^2$ values above 0.80 at the India site while at the urban background site, the PPD60PV had the highest correlation with the tapered element oscillating microbalance (TEOM) with an $R^2$ value of 0.30. At the roadside site, only the PPD20V was used, with an $R^2$ value against the TEOM of 0.18. Emissions factors at the roadside site were calculated as $0.39 \pm 0.10$ g $PM_{2.5}$ per kg fuel and $0.11 \pm 0.01$g BC per kg fuel, which compare well with other studies and estimates based on other instruments. The results of this work show the potential usefulness of these sensors for high concentration applications in developing countries and for their use in generating emissions factors.

## 1 Introduction

Long term exposure to particulate matter (PM), particularly particles less than or equal to 2.5 micrometers in size ($PM_{2.5}$), is associated with a variety of adverse health impacts, including lung cancer (Laden et al., 2006), cardiovascular disease (Laden et al., 2006;Miller et al., 2007;Puett et al., 2009), and premature mortality (Puett et al., 2009). Although some cities in the US have PM values above the National Ambient Air Quality Standard (NAAQS) (EPA, 2013) annual $PM_{2.5}$ concentration value



of 12 µg m$^{-3}$, PM concentrations in many developing countries, including India, are orders of magnitude higher (Tiwari et al., 2015;Health Effects Institute, 2010a).

A variety of instruments are used for PM$_{2.5}$ sampling. The US Federal Reference Method (FRM), is filter-based, non-continuous, and requires skilled personnel and highly specialized facilities and equipment to produce quantitative PM

concentration values (EPA, 2015). Continuous measurement instruments, include US Federal Equivalent methods (FEMs) and other research grade instruments, often cost ten thousand to tens of thousands of dollars and usually need to be operated in climate-controlled spaces, with substantial oversight and maintenance (Chow, 1995). Many PM$_{2.5}$ constituents vary within urban areas (Pinto et al., 2004), but the high costs associated with conventional measurements limit the number of air quality monitoring sites globally, leading to generally sparse spatially-defined air quality information. Citizens and policy makers

desire more data to make decisions for individual and societal health and well-being (Stevens et al., 2014).

Roughly 19% percent of the US population lives near high volume roads (Rowangould, 2013) and in addition, many people are further exposed while commuting on these roads (Greenwald et al., 2014). Emission factors allow researchers to quantify the emissions per unit activity or per unit fuel from vehicles on these roads, providing valuable information to researchers, policy makers, and others. Mobile source emissions continue to be measured around the world due to their importance (Zhang

et al., 1995). A variety of factors can influence vehicle and fleet emissions, including vehicle type and technology, traffic density, and local meteorology (Health Effects Institute, 2010b), and emissions therefore vary over time and regionally (Zhang et al., 1995). The methods that are most widely used to develop emission factors for individual vehicles and fleets include chassis dynamometer testing, portable emissions measurement systems (PEMSs), tunnel studies, and remote sensing techniques (Franco et al., 2013). Although all these methods have advantages and disadvantages, these approaches have high

capital and personnel costs.

New sensor technologies may be able to address some of the issues of cost and convenience posed by conventional measurement equipment. New sensors are available that are lower in cost than their conventional counterparts, down to 1-10% of the cost of a reference analyzer. A further advantage is that these new sensors are small in size, light weight, and have minimal power consumption. These new sensors have been used to identify and monitor hot spots, and in arrays to generate

data with higher spatial resolution (Gao et al., 2015), to collect personal exposure data (Nieuwenhuijsen et al., 2015;Steinle et al., 2015), to collect mobile monitoring data (Bossche et al., 2015), and a variety of other applications, including citizen science. Some sensors have been evaluated in lab conditions in addition to field conditions (Wang et al., 2015;Austin et al., 2015). The new sensors have the potential to be a feasible option for researchers, governments, citizens and community groups to monitor air quality in many more locations. Concerns remain about the accuracy and performance of these newer sensors due to their

lower cost and more simplistic measurement techniques and because they often come with very little information from the manufacturer. This concern can be mitigated by thoroughly evaluating the sensors for specific applications and conditions (Snyder et al., 2013;Kumar et al., 2015).

The goal of this work was to evaluate a variety of lower cost alternatives for generating continuous pollutant measurements. These sensors include several particulate sensors, a CO$_2$ sensor, a black carbon (BC) monitor, and supporting temperature and





humidity sensors. These PM sensors have been deployed both under low (US) and high (India) ambient PM concentration settings. In a novel application, a system of gas-phase and particulate matter (mass and black carbon) sensors was used to calculate in–use emissions factors near a freeway.

## 2 Methods

### 5   2.1 PM Sensor Configurations

This research was conducted primarily through field studies designed to: (i) characterize three commercially available, relatively low-cost optical particle sensors, (ii) develop a sensor measurement package capable of characterizing multiple air pollutants, and (iii) calculate emissions factors using the sensor package. After assembly, the multi-sensor package was applied in multiple field environments to examine how select sensors compared to reference monitors in ambient environments as well

as to derive in situ emission factors along a major roadway. Three low-cost particle sensors were tested (PPD42NS, PPD20V, and PPD60PV, Shinyei Technology Co., Ltd. Kobe, Japan). The Shinyei sensors were selected because of their price and the prevalence of use of the PPD42NS sensor in citizen science applications. These sensors appeared promising after initial evaluation so further testing was performed as detailed in this paper. Details are summarized in Table 1 for all sensors. PM sensors arrived uncalibrated from the manufacturer. The sensors have a 0.25 W resistor that is designed to heat the air drawing

a sample passively into the detection volume. The sensors measure particles using light scattering. An infrared LED is used as the light source, and a photodiode array with lens measures the scattered light at ~45 degrees.

The PPD42NS is a digital sensor: it provides a binary high or low output and sends pulses when particles are detected in the beam. These pulses are summed, and the fraction of time when pulses occur over the total time is calculated. This ratio from the PPD42NS is used to calculate the particle mass by calibrating against a particle mass instrument (either a Tapered Element

Oscillating Microbalance (TEOM) (Thermo Scientific, USA) or an Environmental Beta Attenuation Monitor (E-BAM) (Met One Instruments, Grants Pass, OR, USA). The PPD42NS sensor has problems with stray light penetration due to the open design of the optical sensing chamber. The sensors were placed in opaque junction boxes with no other measures taken to limit light contamination. Previous work compared the Shinyei PPD42NS particle sensor to a variety of reference instruments both at US ambient concentrations (Holstius et al., 2014) and in Xi'an, China, at higher ambient concentrations (Gao et al., 2015).

The other two Shinyei sensors (PPD20V and PPD60PV) have an analog output, with a variable voltage depending on the light scattering occurring in the sensing volume. These sensors also have the capability to function as digital sensors but were not used in this way for our experiments. The manufacturer reports that the PPD42NS and PPD20V detect particles greater than 1µm in size (Shinyei Kaisha, 2002;Shinyei Technology Co., 2010) while the PPD60PV detects particles greater than 0.5 µm in size (Shinyei Technology Co., 2013). Unfortunately, Shinyei provides no further information regarding the design of these

three sensors. All three types of sensors were calibrated in the field using Deming regression (Linnet, 1993) or using an exponential function in the case of non-linear results, using the one hour averages of the sensor and the reference analyzer.



In addition, a low power, 3.5 mW, nondispersive infrared (NDIR) (CO2Meter.com) $CO_2$ sensor was used. The sensor was calibrated for a range of 0-1025 ppm in the laboratory prior to deployment and was recalibrated against a $CO_2$ instrument (Thermo Fisher Scientific Inc. Franklin, MA, USA 410i) in the field using Deming regression of the one hour averages of the sensor and the reference analyzer. In addition, a mid-cost, portable black carbon monitor was added to the sampling package

(microAeth, AE51, Aethlabs, San Francisco, CA, USA). The performance of the microAeth as compared to reference Multiangle Absorption Photometer (MAAP) and Aetholometers has been characterized in previous papers (Cheng and Lin, 2013;Viana et al., 2011). In conjunction with the other sensors, a Sensirion AG (Staefa, Zurich, Switzerland) temperature and relative humidity (RH) sensor (SHT15) that measured temperature by band-gap displacement and RH using a capacitive sensor (Sensiron, 2010) was used.

Other than the microAeth, which has internal data logging, these sensors were wired to an Arduino Mega microcontroller (Arduino, www.arduino.cc, last accessed September 14, 2015) which was paired with a data logging shield (which includes a real-time clock) from Adafruit (New York, NY, USA) that logged the sensor's analog signal or pulse ratio and stored time stamped one-minute averages to comma-separated values (CSVs) on an SD card. These sensors were assembled into plastic junction boxes. Figure 1 shows a 6" x 6" x 4" box with sensors used during the roadside testing. The box and additional

electronics to run these sensors cost just over $100 from a local hardware store and online electronics and electrical box suppliers. A 25 mm fan was positioned to draw air in to the instrument package and was positioned directly below the PM sensor. The air flow volume for the fan as reported by the manufacturer was 67 liters per minute so the exchange rate in the junction box is estimated to be approximately twice per second in the case of the roadside site. The exhaust flowed out the elbow on the right hand wall of the box, and the instrument cables were threaded through the elbow as well.

A slightly different setup was used for the sensor comparison testing where multiple PM sensors were operated at the same time (Figure 2). In this case, three 25 mm fans were positioned to draw air in to the instrument package and again, the exhaust flowed out an elbow. Four of the PM sensors, the three PPD20V sensors and the PPD60PV, were positioned directly above the fans while the PPD42NS was placed on the wall of the box perpendicular to the other sensors as shown in Figure 2. The arrangement of the sensor in the box, may have affected the flow through the PPD42NS but three fans should have provided

ample flow through the PPD42NS and the CO2 and temperature/RH sensors. With three fans, the exchange rate in the junction box was estimated to be approximately six times per second for the comparison box, although possibly less due to flow resistance through the box. We have enclosed the PM sensors and other sensor fans were added to provide additional air flow through the box than the heating resistor alone, since the heating resistor would really only supply flow through the individual PM sensors and not through the whole box. The addition of the fans may change the size of the particles drawn into the sensing

volume, affecting the manufacturer-reported specifications for minimum detectable particle size.





## 2.2 Sensor Evaluation

Particle properties are variable and are composed of both internal and external mixtures of chemical components that vary as a function of size. The response of optically-based PM sensors is largely a function of the actual properties of the ambient aerosol at the measurement location, including the size distribution and chemical composition. Wang et al. (2015) found that the response of optically-based sensors in laboratory tests was highly dependent on composition; varying up to a factor of 10 depending on composition. Austin et al. (2015) found that the response varied by a factor of up to 12 depending upon aerosol size. Therefore this work focuses mainly on field evaluations of sensors, rather than laboratory studies to evaluate sensor response as a function of particle size, composition, and concentration that is not representative of field conditions. However, we do discuss evaluations conducted in our laboratory as well as recent detailed laboratory analyses of similar sensors (Wang et al., 2015;Austin et al., 2015).

### 2.2.1 Field Evaluation: Sampling locations and reference instruments

Sampling locations and dates are detailed in Table 2, which also includes the reference instruments used at each location. The Thermo Scientific Series 1400a TEOM was used as the reference for the two Atlanta sampling periods. The TEOM is a US EPA Federal Equivalent Method (FEM) at a 24-hour averaged level and is used routinely for regulatory and research monitoring (EPA, 2015). A high efficiency particle arresting (HEPA) filter was attached at the inlet on the TEOM periodically to ensure that the instrument was functioning properly. The E-BAM was the primary reference instrument used in Hyderabad, India, and is a more portable monitoring option than a traditional BAM, operating in the environment without requiring an exterior enclosure (Met One Instruments, 2008). The E-BAM is not a registered FEM in the U.S., although the instrument strongly correlates with federal reference methods (USDA Forest Service, 2006) and has been used as a reference instrument in past studies (Ancelet et al., 2012). Periodic leak checks, flow checks, and monthly nozzle/vane cleanings were performed to ensure proper function of the E-BAM.

Measurements from three different sampling locations (an urban background in Atlanta, a roadside in Atlanta, and Hyderabad, India) were analyzed in this study. The first measurement campaign was at the side of the freeway on the Georgia Tech campus, Atlanta, GA, (33.775560, -84.390950), adjacent to a 15-lane freeway with an Average Annual Daily Traffic (AADT) of 293,256 vehicles in 2014 (Interstate 75 & 85) (GDOT, 2014). The sensor box (Figure 1) was mounted onto a pole on top of a trailer approximately 4 meters above ground. The trailer was parked in a lot separated from the highway by only a fence, leaving the sensor package approximately 6 m from the closest lane of traffic. Next, a comparison was performed on the roof top of the Ford Environmental Science and Technology Building, a four-story building on the Georgia Tech campus, approximately 500 m from the freeway (33.779175, -84.395730). The rooftop, urban background site was above the tree level but there were a few structures on the roof such as an indoor roof top laboratory and building air handling equipment. Lastly, the same sensor package that was deployed on the Atlanta roof top was deployed in Hyderabad, India (17.425798, 78.526814). The sensor package was deployed on a roof top at the National Institute of Nutrition (NIN).



The selection of these three sites gives us a variety of concentration ranges to help determine appropriate uses of the sensors. For sensors that showed at least marginal correlation ($R^2 > 0.1$), Deming regression was applied to the one-hour averaged data to convert the raw voltage output from the PM sensors into an estimated $PM_{2.5}$ mass concentration (Table 2). Deming regression accounted for uncertainties in both the sensors and the reference analyzers (Linnet, 1993) as there is variability not only in the

sensors but also in the reference instruments on this shorter one-hour time scale. The $R^2$ values are almost identical when using Deming or linear regression ($\pm$ 0.03). Since reference analyzers were present during the entire time the sensors were being evaluated, the calibration was generated using data from the whole period.

### 2.2.2 Laboratory evaluation

A chamber experiment was also run with the 3 PM sensors. A 284 liter modified sealed glove box with a slight positive pressure

was used. A puff of incense smoke was introduced into the chamber and the concentration was allowed to decay while clean air was pumped into the chamber. Over a 1 hour period the concentration dropped from above 500 to 0 µg m$^{-3}$ As measured by a TSI TSI DustTrak 8533 (Shoreview, MN). The sensors were located inside the chamber while a forked line ran from the chamber with a short length of antistatic tubing going to the DustTrak and the other exhausted through a filter and into the lab. The correlation between the 3 sensors and the DustTrak was compared.

## 3    Results and discussion

Results from these three measurement periods characterize a wide array of atmospheric conditions and different urban surroundings, as well as differences due to source contributions (Table 3). The results have also been compared to results from lab evaluations from this work (Table 4) and previous lab evaluations.

### 3.1    Ambient concentration comparisons

### 3.1.1 Urban Roadside

The first measurement campaign was at the side of the freeway on the Georgia Tech campus. The Shinyei PPD20V sensor was within 4 µg m$^{-3}$ of the TEOM at an hour average with reference readings ranging from fairly low (~10 µg m$^{-3}$) to moderate (maximum of 32 µg m$^{-3}$) concentration levels. Over this three day campaign, the PPD20V at times tracked the TEOM well (e.g., for ~12 hours on 10/2) but at other times showed significant disagreement (e.g., ~20 µg m$^{-3}$ difference on 10/3). In some

cases, not only was there a difference in the magnitude of the response but also a disagreement in the direction of the response. This comparison between the TEOM and the PPD20V provided a low overall correlation of 0.18 (Figure 3).

Although much of this low correlation is due to errors in the Shinyei sensors, some of the inaccuracies may also lie in the TEOM (Allen et al., 1997), especially when using 1-hour versus 24-hour data. The concentrations were low at this location, which may cause a higher relative error in the TEOM since it is measuring very small masses. In addition, the TEOM operates

at a temperature higher than ambient to reduce humidity interferences, but operation at a temperature higher than ambient



causes the loss of some of the semivolatile organic fraction, especially when the temperature outside is much cooler (Tortajada-Genaro and Borras, 2011). Since the Shinyei sensors measure via light-scattering, the associated inaccuracies will be due to the wavelength of the light source, the difference between the actual versus calibration aerosol size distribution and composition, and sensitivity of the detector.

The low cost $CO_2$ sensor compared closely with the reference monitor at an hourly average ($R^2 = 0.75$) over the four-day period (Figure 4). Figure 4 shows the one hour averaged data. The two devices track closely with each other over the three-day time period except for two peaks on 10/3 and a peak near the initiation of testing on 10/1 detected by the reference monitor but not the low cost sensor. During these periods, for unknown reasons the sensor response was ~100 ppm lower than the reference analyzer, and the discrepancy does not appear to be related to extreme temperature or humidity events as the ambient

conditions were very close to those of the day before when data compared better. During this test, 69% of the time the RH was below 70%, with 15% above 80% RH and no data included above 90%.

### 3.1.2 Urban Background

Next, a comparison was performed on the roof top of the Ford Environmental Science and Technology Building. Figure 5 shows the raw signals from the low cost particle sensors compared with the concentrations recorded by the TEOM on a one-

hour average. The concentrations of $PM_{2.5}$ seen on the roof were low (mean: 8 µg m$^{-3}$), and the PPD60PV was the only sensor to achieve an $R^2$ value above 0.1 with an $R^2$ of 0.30 (Figure 6). While the 3 PPD20V sensors do show agreeable precision with high correlations, they do not agree well with the TEOM ($R^2$ 0.1 to 0.0). Therefore, no calibrations were performed between the sensors and the TEOM, allowing no standard errors to be calculated since the raw output of the sensors have no meaning uncalibrated. In their current configuration, all of the low-cost particle sensors had low to no correlation with the TEOM while

measuring lower urban background concentrations. This lack of correlation may be due to not only the poor performance of the sensors but also to the way they were assembled in the junction box. Testing occurred during December during colder weather with 50% of the data being above 70% RH and 38% above 80% RH, likely leading to the large errors associated with this time period.

### 3.1.3 High Ambient Concentrations

Lastly, the same sensor package that was deployed on the Atlanta roof top was deployed on a roof top in Hyderabad, India. The results from India show higher average PM concentrations (1 hour averaged 72 µg m$^{-3}$ range: 8-247 µg m$^{-3}$) over the one month deployment period (Figure 8). The PPD60PV approaches saturation, as indicated by the exponential shape of the comparison where the PPD60PV reported concentration levels off at concentrations above about 100 µg m$^{-3}$ in the Hyderabad environment, so the resulting relationship between the E-BAM and PPD60PV is nonlinear (Figure 7). Gao et al. (Gao et al.,

2015) observed saturation with the PPD42NS sensor functioning at slightly higher concentrations in Xi'an China (hourly E-BAM average of 485 µg m$^{-3}$ range: 77.0-889.0 µg m$^{-3}$). We applied an exponential function to calibrate the PPD60PV against the E-BAM, whereas Gao et al. applied a fifth order polynomial to the PPD42NS signal that included temperature and humidity





terms (Gao et al., 2015). Since the relationship between particle scattering and mass is usually linear (Chow et al., 2002), and the physical meaning of the shape of a fifth order polynomial was not seen for this application, an exponential function was used to represent saturation. The $R^2$ values were very similar between the fifth order polynomial (0.64) and the exponential curve (0.62). An exponential function was applied as a calibration to the raw voltage signal data produced by the sensor,

yielding Figure 8.D. The coefficients of determination were greater than 0.8 for the PPD20Vs in this study. The RH was 70% or below during 70% of the sample period with 15% of the data falling between 80 and 90% RH and no data included above 90% RH. The PPD60PV sensor in India was the only sensor that was calibrated using a non-linear fit since our concentrations were lower.

Simultaneous operation of three PPD20V sensors allowed comparison of sensors of the same type. The three PPD20V sensors

have similar coefficients of determination (0.81-0.86) and standard errors (SEs) (16-20 µg m$^{-3}$) see Figure 8. Also recorded on the graphs are the slopes and intercepts of the calibrations whereas the graphs depict the post-calibration data (slope = 1, intercept = 0). PPD20V sensors 1 and 3 (Figures 8.A and 8.C) have similar calibrations, with intercepts of -56 and -69. The second sensor has an intercept of -115, almost twice that of the other two sensors, and the slightly increased slope suggests that this sensor is slightly less sensitive to changes in PM concentration than the other sensors. Although this is a small sample

size of sensors, these widely differing calibrations show the need for individual calibration for each sensor, even those of the same model. In addition, these calibrations are different from the calibration generated at the roadside (m = 0.32, b = 4.6). Differences are expected in different locations based on different optical properties of the aerosols (Chow et al., 2002), so differences in optical properties such as size or color may be causing different calibrations in different locations.

A variety of reasons exist for the inaccuracies in each concentration range. The India concentrations seem more appropriate

for the PPD20V, while at the low concentrations observed at the background roof site, the sensor agreed poorly with the reference. The PPD60PV agreed most closely with the reference TEOM at the background site, likely because of the ability of this sensor to detect the smaller particles. Additional work must occur to improve the performance of these sensors, especially at the roadside and background sites. However, at the high concentrations in India (1 hour averaged 72 µg m$^{-3}$ range: 8-247 µg m$^{-3}$), the sensor often became saturated. The PPD42NS did not compare well at any concentration observed in this

study. We concluded that light contamination was a factor possibly contributing to its poor performance in this study. Light contamination has been a problem in previous studies as well (Williams, 2014).

A variety of factors affect light scattering, including particle size, shape, composition and relative humidity. The relationship between mass and light scattering is often highly correlated, but the relationship may be different in different locations and during different times of the year. This difference in the relationship has been shown in previous work using nephelometers

(Chow et al., 2002). Comparisons for PM$_{2.5}$ mass and light scattering with nephelometers are usually done using only the fine size fraction and under dry conditions, where the sample is heated to decrease RH to provide the most accurate results (Chow et al., 2002). Our study was done under ambient conditions, and we did not separate the smaller size fraction. Adding in a size separation device, to remove particles greater than 2.5 µm, would have been prohibitively expensive for the low cost setup we were trying to design since without it, we were able to use a fan instead of a pump, thereby drastically lowering the cost, size,



and power consumption of our device. In previous studies, total scattering has been compared with $PM_{2.5}$ mass yielding linear relationships with high $R^2$ values ($\geq 0.9$) (Watson et al., 1991; Chow et al., 2002; Doran et al., 1998). Previous studies using low cost (\$150-\$2050) scattering PM sensors have not performed as well at US ambient concentrations (~0-30 µg m$^{-3}$) with max $R^2$ with FEMs of 0.8 (Williams, 2014). So it is possible the sensors could perform better in future studies in an improved

enclosure.

In previous studies, dried and undried $PM_{2.5}$ light scattering has been compared with the effects of RH and has been relatively constant until 80% RH, with increasing errors after 80%. The largest errors occurred above 90% (Chow et al., 2002). Other studies have shown more dependence on RH, even at lower values with differences in scattering coefficients seen between 50 and 70%. The growth of the particles and therefore the scattering of the particles is variable in different locations and over

time as the composition of the particles is different, leading to more or less water uptake (Day and Malm, 2001). In addition, the manufacturer reports that the operating humidity range should stay at 95% or less (Shinyei Kaisha, 2002; Shinyei Technology Co., 2010, 2013). During all three tests, the RH in the sensing box was never above this value and was rarely above 90%, so no RH correction was applied. The temperature range reported by the manufacturer is 0-45 °C (Shinyei Technology Co., 2010; Shinyei Kaisha, 2002; Shinyei Technology Co., 2013), which was also not exceeded during testing.

Although the temperature and humidity never exceeded the manufacturer-specified operation range, there could still be some dependence on RH and temperature as shown by Gao et al. in high concentration environments (Gao et al., 2015) and also by Williams et al. in US lower concentrations environments (Williams, 2014). The electronics may be affected by temperature, as increased temperature increases the resistance in electronic circuits, which could affect the analog sensors.

### 3.1.4  Laboratory Comparison

During the chamber experiment the performance of the 3 Shinyei sensors were evaluated by comparison with a DustTrak monitor. One minute averaged data was analyzed and the PPD20V provided the best correlation over all concentration ranges ($R^2= 0.70$ from 0-50 µg m$^{-3}$ and $R^2= 0.99$ from 0-500 µg m$^{-3}$) (Table 5). The PPD20V also showed the best correlation during the field tests.

The PPD60PV performed poorly at low concentrations in the lab ($R^2= 0.20$ for 1 minute averages from 0-50 µg m$^{-3}$ in the lab).

This was also seen in the ambient work on the roof top at similar conditions ($R^2= 0.30$ for 1 hour averages from 0-38 µg m$^{-3}$ on the rooftop). At higher concentrations the coefficient of determination was higher ($R^2= 0.87$ 0-500 µg m$^{-3}$) and the saturation that occurred during the India field experiments was not seen. This is likely due to the difference in chemical composition, mixing state and size distribution for the lab generated incense versus in the ambient particulate matter in India.

Better accuracy has been reported for the PPD42NS by Austin et al. 2015 ($R^2=0.66$-0.99 depending on particle diameter from

0-50 µg m$^{-3}$) than was seen in our lab results ($R^2= 0.20$ from 0-50 µg m$^{-3}$). Wang et al. also reported much higher $R^2$ in their lab calibrations with incense ($R^2= 0.95$ from 0-100 µg m$^{-3}$). This may be due to longer rolling averaging times (Austin et al., 2015), longer sampling time (Wang et al., 2015) differences in microcontroller signal processing, and difference in comparison instruments (Austin et al., 2015; Wang et al., 2015).





The limit of detection (LOD) was also calculated in the lab by using the 95% confidence interval of the intercept after lab calibration on the 0-100 µg m$^{-3}$ range. The PPD42NS LOD calculated was 9.1 µg m$^{-3}$ which is higher than measured in Wang et al. (2015) (4.59 µg m$^{-3}$). The PPD20V has an LOD of 4.6 µg m$^{-3}$ while the PPD60PV has a higher limit of detection of 29 µg m$^{-3}$.

The challenge with optically-based PM sensors is that the actual response (i.e., sensor calibration) is largely a function of the actual properties of the ambient aerosol at the measurement location, including the size distribution and chemical composition. Further, the relationship can depend upon composition-related optical properties, and would also be RH dependent (Chow et al., 2002;Wang et al., 2015). Calibration to a mono- or poly-disperse calibration aerosol of a specific aerosol (e.g., sulfate or polystyrene latex (psl)) or to another particle source such as incense, can lead to biases as the actual response in the field can

be significantly different (Dacunto et al., 2015;Jiang et al., 2011;Wang et al., 2015;Austin et al., 2015). It is likely that the response to a laboratory generated aerosol will be much different to that in the field. In the end, calibration should be done with an aerosol similar to that being sampled. This is especially important given that the focus of this paper is on the use of an inexpensive sensor package to estimate emission factors where the focus is on the change in the local PM levels associated with changes in related gases. The dependence on aerosol properties will also impact the LOD as aerosol properties can be

associated with concentration (e.g., periods of high concentrations will have different composition than at low concentrations).

### 3.2 Estimating Vehicle Emissions Factors

The instrument package, as part of the roadside deployment, was used to quantify both PM$_{2.5}$ and BC emissions factors (EFs) for traffic on the freeway. Calculating emissions factors normalizes the pollutants by fuel consumption (i.e., grams of pollutant per kg fuel burned). In this case, the emissions of the entire fleet of cars on the road during the sample periods were calculated

rather than generating emissions factors specific to certain vehicles as has been done in some remote sensing applications. In this study, a carbon balance method was used to calculate the EFs (Singer and Harley, 1996) (Eq. (1)):

$$\text{EF} = \left( \frac{[P]}{\sum[C]} \right) * \frac{w_c \rho_f}{MW_c}$$

(1)

where $[P]$ is the concentration of the pollutant of interest in the exhaust, $\sum[C]$ is the sum of the carbon species (CO$_2$, CO, and

hydrocarbons), $w_c$ is the weight fraction of carbon in the fuel, $\rho_f$ is the density of the fuel, and $MW_c$ is the molecular weight of carbon (12 g mol$^{-1}$).

In this study, the CO$_2$ was converted into a carbon based fuel based on the assumption that an overwhelming majority of the carbon in the fuel forms CO$_2$ (IPCC, 2006). To convert from volume CO$_2$ to mass CO$_2$, we used 20 °C and 1 atm, in line with observations during that period. Since these data were collected in October of 2013 in Atlanta, the fuel burned on the highway

was assumed to be reformulated winter gasoline (EPA, 2014) with a composition of 86.05% kg C/kg fuel (US Government, 2009). There is some error in these estimates since some of the fuel burned on the freeway is diesel and a very small minority



of other fuels (compressed natural gas (CNG), liquefied natural gas (LNG), etc.) but the carbon content of diesel fuel is very near gasoline (0.8647 kg C/kg fuel (US Government, 2009), so the presence of diesel vehicles will have a minimal impact on the assumed fuel carbon content. Further, this section of the freeway is not open to heavy duty vehicles except those making local deliveries, so the freeway is dominated by gasoline-fueled vehicles.

Emissions factors for the freeway fleet were calculated using changes in PM over temporally-matched changes in $CO_2$. Events were selected where both the pollutant and $CO_2$ rose and fell at the same time. The changes were calculated by subtracting the concentration at the beginning of the event (assumed baseline or background concentration) from the concentrations during the event. This concentration at the beginning of the event may not be a true background concentration as there are still cars on the roads outside these peak times, but we could not integrate from zero since this "background" includes not only some

vehicle emissions but also likely contributions from the regional background and possibly other nearby sources. Therefore, the concentration at the beginning of the event was used as a basis for these calculations and by integrating the excess concentrations of $PM_{2.5}$, BC and $CO_2$ above the "background" level, the emissions factors were generated. This method is similar to the method that has been used in multiple previous studies (Galvis et al., 2013;Klems et al., 2011). However, in this case, we did not have wind direction data to factor into our calculations, so it is an over simplified case.

The data were averaged to five minute means to provide a more stable background at the first point of the period. $PM_{2.5}$ emissions factors were calculated using results from the morning rush hour period, 7:20-8:40 AM (Figure 9A). This time period was selected because we know it is approximately the time of the morning rush hour and because both PM and $CO_2$ rise and fall together over this period. The resulting emissions factor was 0.39 ± 0.10 g PM per kg fuel, between the gasoline (0.038 g $PM_{2.5}$ per kg fuel) and diesel emissions factors (1.4 g $PM_{2.5}$ per kg fuel) derived in other studies (Ban-Weiss et al.,

2008;Dallmann et al., 2013). The sensor error was calculated based on propagation of errors of the SE between the 1-hour sensor and the reference analyzer concentrations. Based on the previously reported gasoline and diesel values, this result suggests that 30% of the fuel on the freeway being burned is diesel, which is likely high. Since the TEOM data were collected in 1 hour intervals over the period, no emissions factor was calculated since the time period would not overlap, introducing error into the comparison.

The area under the BC curve was integrated from where elevated concentrations of both $CO_2$ and BC occurred (Figure 9b). The MicroAeth worked for only the first 12 hours of this test, so the same morning time period could not be used as the PM data. During this period, the BC emissions factor was calculated to be 0.11 ± 0.01 g BC per kg fuel, very similar to the emissions factor that was calculated from the reference instruments (MAAP for BC, TS 410i for $CO_2$) of 0.13 g BC per kg fuel using the same time series analysis procedure. This value is between the gasoline (0.010 g BC per kg fuel) and diesel

emissions (0.92 g BC per kg fuel) estimated in previous studies (Ban-Weiss et al., 2008;Dallmann et al., 2013). The sensor EF suggests approximately 13% of the fuel burned during this period was diesel based on the previously reported values (15% from the reference instruments). Table 4 shows a summary of the emissions factors calculated. In addition, the good correlation seen from the PM sensors in India (PPD20V $R^2$ ≥0.8) might suggest that this method could be used to calculate emissions factors in different environments, as well.




## 4 Conclusions

Of the sensors used in this study, the PPD60PV was most highly correlated with the TEOM at the ambient concentrations observed from the roof top (average: 8 μg m$^{-3}$ R$^2$ = 0.30), and the PPD20Vs were better correlated with the E-BAM in India
(average: 72 μg m$^{-3}$ R$^2$≥0.81). Additional modifications such as light shields and temperature or humidity corrections might improve this sensor's performance. The few week deployment in India under fairly high concentrations without any optics maintenance suggests that these sensors could be deployed a few weeks or more under these conditions, and even longer in lower concentration locations.

These sensors suggest a means to generate in-use emissions factors across a large range in environments at 1 to 2 orders of
magnitude less cost than conventional methods. This cost savings is particularly exciting since emissions factors vary based on the source, and quantifying vehicle fleet emissions factors can be challenging since fleets vary regionally and over time. This type of low-cost sensor could allow for emissions factors to be measured in many more places, providing information for use in air quality studies and to better protect public health. Efforts to improve the accuracy, including characterizing the RH dependence, of these sensors so that these emissions factors are more accurate should continue. Additional characterization
under different environmental conditions and differently polluted areas would also be beneficial.

### Acknowledgements

This work was made possible by the NSF PIRE grant 1243535 and EPA Star grant R83503901. Thanks to J. Jeyaraman, R. Weber, L. King, and J. Hu at Georgia Tech and to J. Marshall at the University of Minnesota. The contents of this paper are solely the responsibility of the grantee and do not necessarily represent the official views of the US EPA or NSF. Further,
US EPA and NSF do not endorse the purchase of any commercial products or services mentioned in this paper. Although an EPA employee contributed to this article, the research presented was not performed or funded by and was not subject to EPA's quality system requirements. Consequently, the views, interpretations, and conclusions expressed in the article are solely those of the authors and do not necessarily reflect or represent EPA's views or policies.

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

**Table 1: Sensors used**

| Pollutant | Sensor | Cost ($) | Technology |
|---|---|---|---|
| PM | Shinyei PPD42NS | 10 | volume light scattering (digital output) |
| PM | Shinyei PPD20V | 250 | volume light scattering (analog output) |
| PM | Shinyei PPD60PV | 250 | volume light scattering (analog output) |
| $CO_2$ | COZIR GC-0010 | 120 | non-dispersive infrared absorption |
| Temperature and RH | Sensirion SHT 15 | 40 | band-gap displacement capacitance |
| BC | Aethlabs AE51 | 6000 | filter absorbance change |





**Table 2: Sample locations and dates**

| Date | Location | Reference | Sensor Model |
|---|---|---|---|
| 10/1/13-10/4/13 | Atlanta Roadside (33.775560, 84.390950), | TEOM | Shinyei PPD20V |
| | | Thermo Scientific 410i | COZIR |
| | | MAAP | Microaeth |
| 11/21/13-12/16/13 | Atlanta Roof Top (33.779175, 84.395730) | TEOM | Shinyei PPD42NS<br>Shinyei PPD20V (x3)<br>Shinyei PPD60PV |
| 1/30/14-2/10/14 | Hyderabad, India (17.425798, 78.526814) | E-BAM | Shinyei PPD42NS<br>Shinyei PPD20V (x3)<br>Shinyei PPD60PV |

**Table 3: Results from comparison between PM sensors and reference instruments during deployments in India and Atlanta**

| Location (Reference instrument) | 1-h Reference Concentration Range ($\mu g\ m^{-3}$) | 1-h Reference Average Concentration ($\mu g\ m^{-3}$) | Temperature and RH Range (°C, %) | PM Sensor Model (Shinyei) | $R^2$ | Standard Error (SE) ($\mu g\ m^{-3}$) |
|---|---|---|---|---|---|---|
| Atlanta Roadside (TEOM) | 10-32 | 21 | 18-35 | PPD20V | 0.18 | 8 |
| | | | 30-89 | | | |
| Atlanta Rooftop-urban background (TEOM) | 0.5-38 | 8 | 0-27 | PPD42NS | 0.02 | N/A[b] |
| | | | 13-92 | PPD20V 1 | 0.00 | N/A |
| | | | | PPD20V 2 | 0.09 | N/A |
| | | | | PPD20V 3 | 0.00 | N/A |
| | | | | PPD60PV | 0.30 | 17 |
| Hyderabad (E-BAM) | 8-247 | 72 | 18-41 | PPD42NS | 0.10 | 150 |
| | | | 13-91 | PPD20V 1 | 0.83 | 18 |
| | | | | PPD20V 2 | 0.81 | 20 |
| | | | | PPD20V 3 | 0.86 | 16 |
| | | | | PPD60PV[a] | 0.59 | 37 |

5   [a]Raw signal fit with exponential curve

[b]Standard Error N/A for sensors with correlations <0.10 where calibration was not generated



**Table 4: Atlanta roadside emissions factors estimates**

| Study | Source | PM$_{2.5}$ (g kg$^{-1}$) | BC (g kg$^{-1}$) |
|---|---|---|---|
| (Ban-Weiss et al., 2008) | Comparison: Mid-Duty and Heavy Duty Diesel | 1.4 ± 0.3 | 0.92 ± 0.07 |
| (Dallmann et al., 2013) | Comparison: Light Duty Gasoline | 0.038 ± 0.010 | 0.010 ± 0.002 |
| Sensors this study | Atlanta roadside | 0.39 ±0.10[a] | 0.11 ±0.01[a] |
| Reference analyzers this study | Atlanta roadside | N/A | 0.13 |

[a]Error based on propagation of errors using 1-hour standard error between sensor and analyzer estimates

5   **Table 5: Laboratory coefficient of determination with TSI DustTrak using puff of incense smoke in chamber**

| Sensor R$^2$ | Concentration range (µg m$^{-3}$) | | | | Limit of Detection (µg m$^{-3}$) |
|---|---|---|---|---|---|
| | 0-500 | 0-200 | 0-100 | 0-50 | |
| PPD42NS | 0.80 | 0.73 | 0.54 | 0.20 | 9.1 |
| PPD20V | 0.98 | 0.94 | 0.85 | 0.70 | 4.6 |
| PPD60PV | 0.87 | 0.49 | 0.10 | 0.04 | 29 |

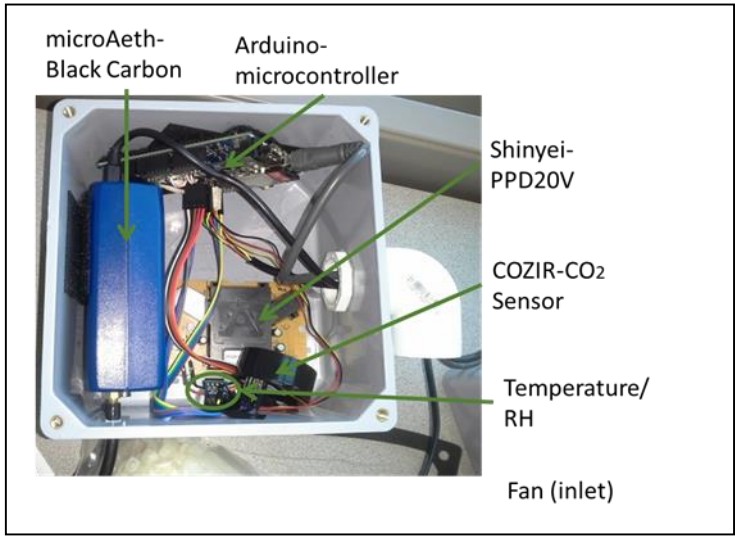

**Figure 1: Sensor Box design used during roadside emissions factor testing**





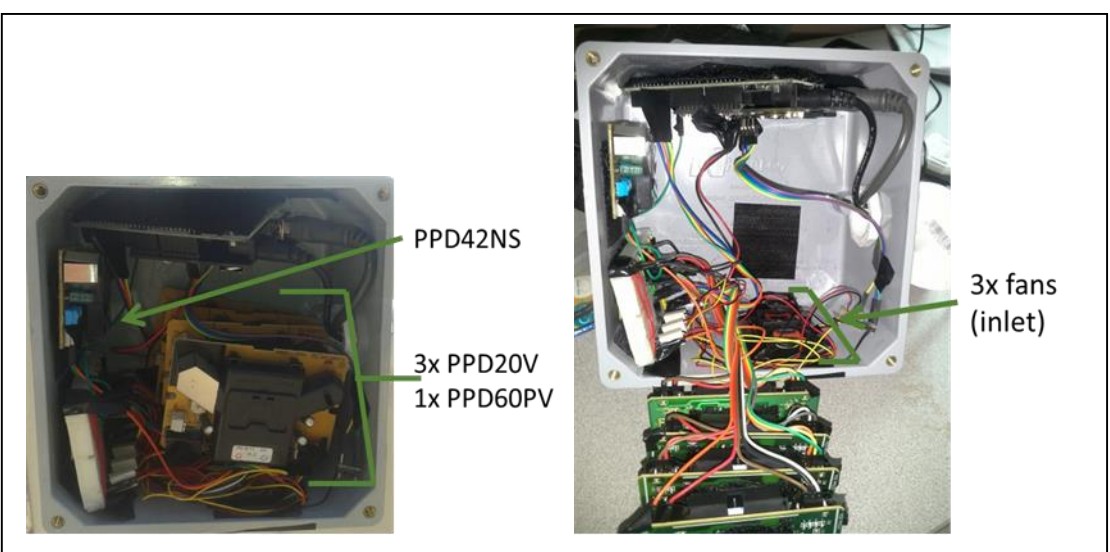

**Figure 2: Shinyei particle sensor comparison box used during Hyderabad, India, and Atlanta rooftop testing.**

**Figure 3: Roadside TEOM, Shinyei PPD20V comparison**





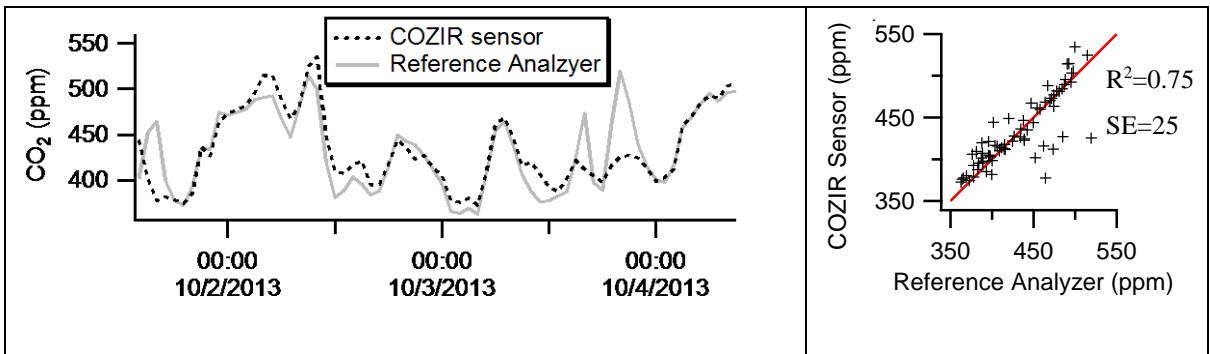

**Figure 4: COZIR reference comparison**

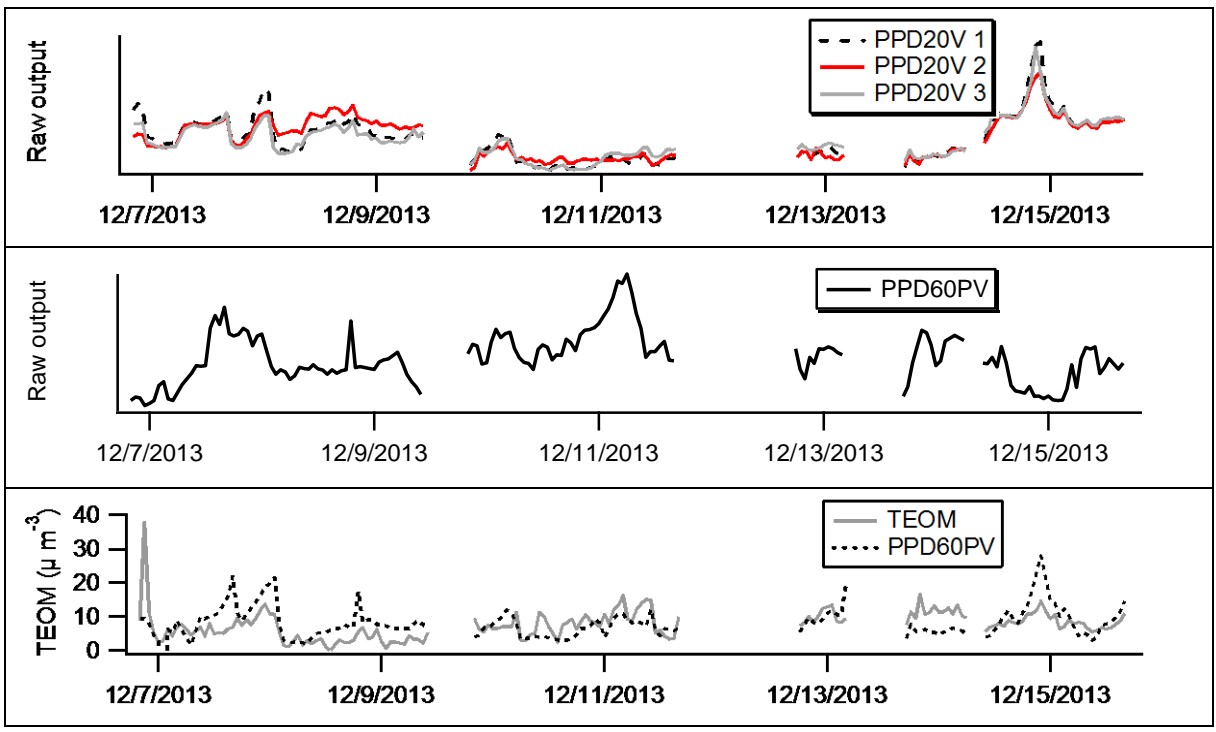

**Figure 5: Rooftop comparison (portion of the full time series analyzed)**



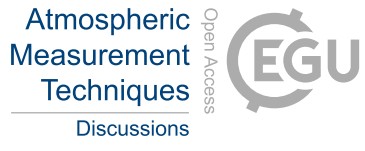

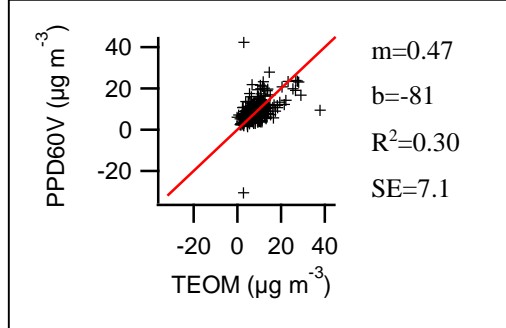

**Figure 6: Rooftop comparison: PPD60PV with linear calibration values**

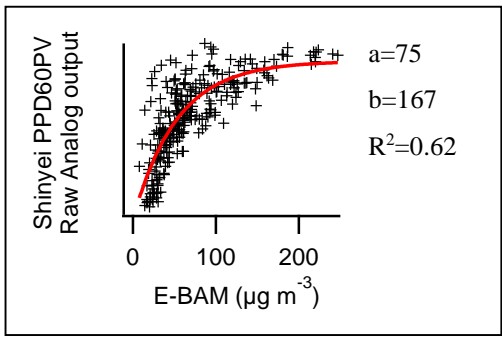

5 **Figure 7: Raw PPD60PV output-exponential fit (Shinyei=a*ln(E-BAM)+b)**



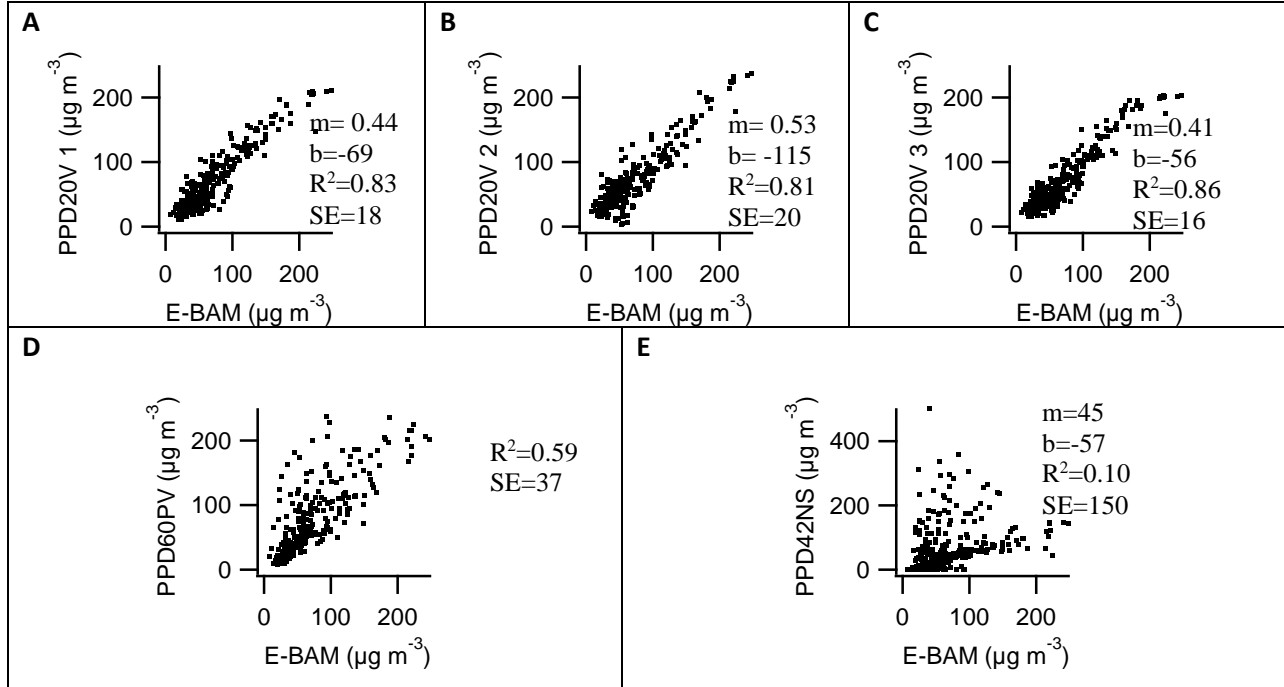

**Figure 8: Results from sensor deployment in Hyderabad, India: E-BAM=m*Shinyei+b**



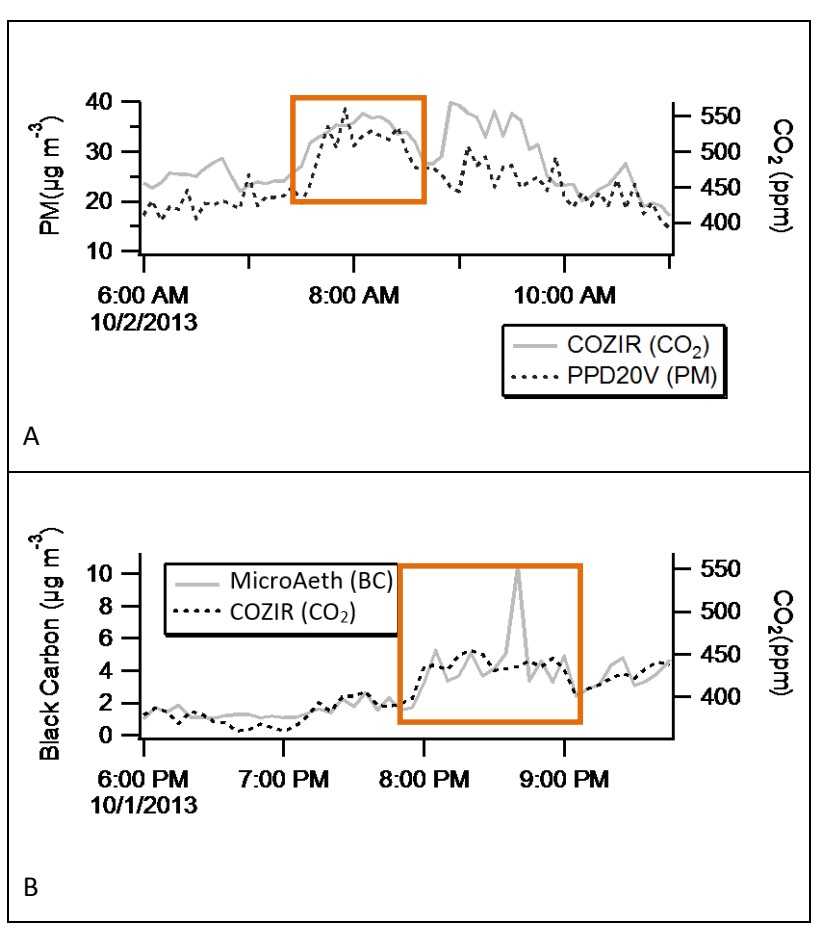

**Figure 9: Emission factors from Atlanta roadside five-minute averaged data (period in orange box where concentrations rise and fall together integrated to calculate emission factors)**

