# Peer review of "Using Low Cost Sensors to Measure Ambient Particulate Matter Concentrations and On-Road Emissions Factors"

_Atmospheric Measurement Techniques, 2015_

## Referee Comment (RC1) · Anonymous Referee #3 · 22 Feb 2016

General comments:

The subject area of this study, low cost sensors for air quality measurements, is a very hot topic. It is therefore important papers in this area are written with great care to understand what new sensors can or cannot do, and a detailed analysis of the measurement data to check for cross-correlation and challenges. Unfortunately this paper has not realised this. The stated goal of the work was: "to evaluate a variety of lower cost alternatives for generating continuous pollutant measurements". However deploying the home-made box of sensors for between a few days up to a few weeks without replicates or complete data analysis of all the parameters is not particularly useful. I would suggest the authors re-analysis all the data which they have recorded

and undertake some studies, e.g. if you used 50% of the measurement period to calibrate the PM sensor, how well does it do against the other 50% of the dataset etc. Some further quality control experiments in the laboratory should also be done before re-submitting to peer review. The authors need to focus less on correlation plots and spend more time on the actual data, and the physical reasons for them, then more may be learned about how to do low cost measurements well.

Specific comments: Abstract The abstract almost wholly misrepresents the results of the study. Rather than reporting poor correlation of the sensors against the reference instrument, that area is almost completely ignored with a focus on emission factors. That part of the paper used less than two hours of data (with a correlation of 0.18 to a reference instrument) to conclude that emission factors could be measured with ∼30% error. The conclusion that the paper's results has showed the potential usefulness of low cost sensors for high concentrations is at best hopeful. The best sensor showed saturation at higher concentrations of PM. I would suggest the abstract be re-written to reflect the results presented.

P3 line 6 Section 2.1 There is a list of aims (i) –(iii). It is not clear having read the whole manuscript if (i) was done at all – the uncalibrated sensors appear to have been installed directly into "a sensor measurement package" (or box). Could the authors specific what (if anything) was done other than directly deploy the sensors and calibrate by running in parallel with other instruments?

Line 12 "The sensors appear promising after initial evaluation" Could this be referenced or detailed.

Line 13 and Table 1: The cost and manufacturer are not really the pertinent details required to evaluate a performance. Please could the authors add to the table summaries of the manufacturer specifications for the sensors . Line13-14 and p6 line 1-7. Calibration: They state in line 30 a Deming regression is used to calibrate. Could they explain why they use this particular regression. The Deming regression assumes the errors for

the two variables are independent (this is reasonable) and normally distributed (this is not known), and the ratio of their variances is known. (it is not known).

The authors are not calibrating the sensors, they are using the reference instrument to calculate a PM from the sensor voltage. And they have assessed the performance by looking at the correlation of the sensor output against the reference TEOM for the same dataset. It would be more standard to calibrating with the TEOM and then comparing it against the TEOM for a different period – which is a reasonable experiment as long as calibration and measurement periods are clearly separated. IF they have done this, they have not communicated it in the manuscript. A flow diagram of their calibration and measurements would help.

P4 line 9: as per previous comment on Table 1, details of the T and RH sensor specifications should be included.

P4 line 17: the authors state that there is a flow of 67 l.m-1. An explanation of why such a high flow rate is used would be useful. What is the response time of the diffusive sampling sensor and then how does that related to the air mass sampled. It would be useful to see a flow diagram with theoretical response time (and empirical from the field measurements). I am slightly concerned that although the sensors are meant to diffusionally sample with the flow rates used, there would be a significant pressure differential on the inlet of the sensor volume and hence they would not be just diffusionally sampling. Is the internal box temperature equal to the external temperature in the field tests?

Line 20 In the multisensor unit the authors add 2 more fans: does this lead to a flow of 201 l.m-1? What effect might that have on the system performance? They note that the configuration means that the PPD42NS is in a different position and hence that may affect the results, but then do not mention it again even though PPD42NS is the worst performing of all the sensors AND (according to the introduction) most widely used. It seems little care was taken when trying to make sure an equivalence was ensured in

the testing.

Line 27 this sentence does not make sense.

Line 29: "the addition of fans may change the size of particles drawn into the sensing volume": Did it? Were any trials of switching the fan on and off or box with and without lid done either in the lab or field?

P5 Section 2.2: This purpose of this paragraphs is difficult to understand. To paraphrase, the authors are saying: lab experiments have shown that different aerosol types give different results, so we assume they are not representative of field conditions so we decided to do a field experiment. Ambient aerosol is mixed! I am not sure whether the authors know what they are doing in this paragraph apart from imply that the lab experiments were not useful. . P 5 line 23: Was the TEOM inlet co-located with the sensor pack at 4 m. Did the TEOM have a PM10 or other size selecting head on it? If the inlets we not co-located what was the separation? There are strong gradients near roadsides therefore if you are testing one method against another these details should be noted. Similarly for the other two locations: Where is the TEOM or eBAM inlet c.f. the sensor package?

P6 line 9: Laboratory evaluation: The authors should either take this out or describe the experimental results fully. A table of the statistics from the experiment is not sufficient to describe the results. How did the three sensors co-vary with the DustTrak? A plot of the DustTrak concentration and the three sensors during the one hour experiment is the minimum needed for the reader to understand the outcomes. It does matter whether there is an offset, whether there is a response lag or if they all co-vary nicely.

Results section: P6 line 28: inaccuracies are attributed to the TEOM when "using1 hour versus 24 hour averages". This needs explaining in more detail. The authors are not using 24 hour data. The uncertainties in the TEOM are real and well characterised by many papers more recent than Allen et al., however stating they exist and looking at the dataset being assessed and working out the level of errors likely are two different

things. The particle masses being measured at the roadside are not very small– small is remote background sites where the mass is 0-5ug.m-3, not roadside.

P7: lines 1- 11. Because the authors have not included a plot which shows the variation of T, RH and the PM or CO2 in one figure it is almost impossible as a reviewer to comment on their statistics. They should have a figure with the data in it and then analyse it openly. Again, how close were the two CO2 measurements? Were they exactly co-located?

Line 16: only the data from the PPD60V is shown in Figure 6. It would be useful to see all the data, despite the poor correlations.

Line 21. The authors imply in this sentence that if the RH>70% large errors are to be expected. Given the sensors are being proposed as ambient PM monitors and ambient RH is frequently above 70% RH, does this not imply the sensors are not fit for the purpose intended? The authors do not seem to have an accurate understanding of ambient atmospheric conditions and also do not spend any discussing this rather large potential problem.

Line 30 onwards: The effect of sensor saturation is discussed only in terms of what curve to fit, rather than trying to understand why it is saturating and whether it is quantitative to use the data outside of the linear region. I suspect that you have multiple scatters and absorbers of light leading to a lower recorded concentration.

P8 and Figure 8: A high correlation coefficient is reported for the PPD20Vs at the Indian high PM concentration site. However inspecting the x-y plot, there is a lot of scatter in the <100 ug.m-3 part of the graphs. Again it is a shame that the time series are not shown as only then can you really see the detail. Unfortunately the scatter plot does not tell a reader enough about the performance. The interesting information will be in the details which the authors have not shown or discussed. Again the authors mention the 70% RH threshold as being important, however I would re-iterate that any sensor used in the environment must be able to function across the RH range.

[Figure]

P9 line 4: The statement that the sensors could perform better in future studies in an improved enclosure has no basis (though may possibly be correct). Why do the authors think this? What proposals for improvements are they proposing?

Line 17: Could the authors measure the temperature inside and outside of the box to address the question of the T variation of the electronics?

P10 line 4: Could the LOD as calculated by 2 or 3 SD of the noise on the blank signal be reported. I think this would be a better standard . Also blanks as a function of temperature and RH would be interesting to see. Is there any baseline drift on the sensors over the measurement periods?

Section 3.2: This is one of the most detailed sections of the paper, however in fact the authors only used just over 1 hours data out of 3 days deployment at the kerb-side, ignored periods when CO2 and PM did no correlate and came up with very good sounding numbers with high certainty (compared to all the other measurements in the paper). Unfortunately I do not feel the numbers are robust enough to publish, based on 1 hours worth of data with no replicates.

Section 4 Conclusions: The conclusions section seems to ignore all the poor statistical performances of the sensors, the potential environmental limitations, the issues with lack of correlation with reference instruments, the unknowns about sensor housing performance, the poor performances at <100ug,m-3 and above 200 ug,m-3 and paints a very rosy summary. This section should be re-written to actually reflect the results reported.

---

## Referee Comment (RC2) · Anonymous Referee #4 · 24 Feb 2016

This manuscript aims to assess the reliability of 3 low-cost Shinyei-brand PM sensors via comparisons with two other PM measurement methods, a TEOM and an E-BAM, for 3 urban locations. While not clearly explained in the mspt, the output from the Shinyei sensors apparently is uncalibrated ("raw"), so sensor reliability was assessed based on how strong the correlation was between the Shinyei's raw output and the reference monitor's reported mass concentration. For a small set of lab experiments, the Shinyei sensors were compared with a DustTrak.

The authors also utilized a low-cost $CO_2$ sensor (COZIR GC-0010), showing a comparison with a reference monitor for one location. While a "mid-cost" MicroAeth BC sensor was also deployed, no comparisons to reference monitors were made in this

study.

Characterizing the performance of new monitoring devices under field conditions is a worthwhile goal. However, the number of sites measured (3), the small range of concentrations seen at 2 of the sites, and the analytical insights seem insufficient to contribute to this goal. The only insight this reader gained from the manuscript is that the Shinyei sensors are unreliable – the correlation with reference monitors is much too low to view the devices as even semi-quantitative. In addition, given that they fail to capture temporal trends some of the time (p.6, line 25), they cannot be viewed as qualitatively reliable, either. I am not sure if this modest insight (which was already known for one of the models), by itself, merits publication. If it does, then there are a number of other major issues needing to be addressed, as itemized below.

Major issues:

1) The manufacturer reports different lower particle size detection limits for different models (ranging from 0.5 – 1 $\mu$m; page 3, line 28). For a motor vehicle-dominated site, one would expect much of the mass to be submicron – this would lead to differences in measurements for the different models, as well as with the TEOM (which uses a filter with a very high capture efficiency for submicron particles). The mspt needs to discuss how/why a sensor that only detects down to 1 $\mu$m would be effective at estimating EF values for combustion emissions.

2) While I did not see this mentioned until p.8 (line 33), it appears that the Shinyei sensors were measuring total PM, with no size cut. I presume that the E-BAM was operated with a PM2.5 inlet, although this is never stated in the mspt. However, I believe the model 1400a TEOM is designed for PM10 measurements. The authors need to explain why the plots, and much of the results/discussion, list PM2.5 as what is being measured or estimated.

3) The discussion of biases in the TEOM measurements seems inadequate (pp. 6-7). There have been quite a few papers since Allen et al's 1997 paper comparing the

TEOM with other sensors, including those relying on light scattering measurements. As just one example, Karagulian et al (JEM 14:2145, 2012) found an R2 value of 0.75 between the TEOM and a SidePak. It would seem important to note how well previous comparisons between TEOMs and other light scattering instruments have worked, as context for (and comparison with) the measurements from the Shinyei sensors.

4) How does the level of air pre-heating for the Shinyei sensors compare with that for the TEOM? While heating in the TEOM is mentioned as a possible artifact on p.6 (line 30), there is no mention at this point in the mspt that the Shinyei sensors are also heated (even though this was stated back on p.3 line 14).

5) How did the timing of disagreements between PM monitors (p. 6, line 24) relate to the RH measurements (or the temperature measurements)?

6) For the Hyderabad measurements, clearly the R2 value is driven by the highest concentration data points. If only values <40 $\mu$g/m3 are used, does the R2 become similar to the Atlanta rooftop?

7) The entire "Estimating emission factors" section is inadequately supported, and should be omitted, due to the following concerns:

a. The R2 for this site (between PPD20PV and the TEOM) was 0.18, which is a very uncertain starting point for relying on the PPD20PV values in calculations.

b. As is mentioned in this revised version of the mspt, the light scattering characteristics of the PM will vary between "background urban" and vehicle emissions. So applying a single regression to the Atlanta roadside, where only a subset of the data is believe to be vehicle-dominated, will lead to even more inaccuracies.

c. The EF estimate is based on a 5-min period of elevated PM and CO2 data, even though 1-min measurements were collected over 3 days. Oddly, looking at the CO2 data (Figure 4), it appears that the 5-min period chosen is in the midst of a ~6 hr period where CO2 measurements were ~480-550ppm. But the synchronous PPD20V

data (Figure 3) show PM levels fluctuating much more frequently over this same time period.

d. No evidence is provided that the approach to choosing this 5-min period involved objective (=statistical) analyses – the authors only say (p.11, line 17) that they chose a morning rush hour (even though there would have been 3 morning rush hours of data), "where both the pollutant and CO2 rose and fell at the same time". Why weren't more time periods tested?

e. The lack of information on wind direction, traffic density, and "background" concentration levels (upwind of the roadway) leads to more uncertainties and unsupported assumptions.

f. After obtaining an EF value, and deducing that this would correspond to about 30% of the fuel combustion involving diesel, on a freeway that should have been "dominated by gasoline-fueled vehicles" (p.11, line 4), the mspt says (p.11, line 22) that this EF value "is likely high".

g. Then, a 2nd EF is found, based on BC and involving a different time period (8-9pm), with no justification for why this period was expected to have high concentrations of roadway emissions. This EF corresponds to 13% diesel combustion – but there is no discussion of whether this EF value is trustworthy or believable. Is it feasible that there could be this large a portion of diesel vehicles making local deliveries, in the evening?

h. I am extremely skeptical that the confidence bounds shown with the EF values are accurate – they seem much too low, and the method used to quantify uncertainty isn't clearly explained. By SE, are you giving the standard error of estimate, representing the 85% (that is, 1 sigma) prediction interval around each best fit line? But how are you accounting for the fact that, for the roadside site, only 18% of the variability in the Shinyei measurements can be accounted for by the reference PM measurements?

8) On p.14, it is noted that the optics weren't maintained at all during the "few week"

deployment in India. How do the authors know that the sensors were still performing acceptably near the end of the deployment period? No temporal measurements are shown or were discussed for the India deployment, nor were tests performed to assess how frequently optics maintenance was needed.

9) The use of an exponential (monotonically increasing) eqn to capture a saturation-type effect seems misleading – it gives the erroneous impression that, despite saturation, an accurate mass concentration can still be inferred. If saturation was indeed a problem, then all measurements greater than a certain raw output level should have been omitted from the fitting protocol, and subsequent analyses.

10) The exact same set of data used to determine the best fit line (or exponential) was then transformed, using this best fit equation. It does not seem scientifically appropriate to apply a transformation equation to the exact same set of data that was used to find the equation.

11) In Figure 5, there are noticeable discrepancies between the "raw output" data for the PPD60PV and the $\mu$g/m3. In some cases, e.g., around the 12/11 and 12/13 tick marks, large variations in the raw data appear to have been almost completely smoothed out in the $\mu$g/m3 plot. In other cases, e.g., around the 12/15 tickmark, a large spike has appeared in the $\mu$g/m3 plot that is completely absent in the raw output plot. Given the monotonic nature of the fitting equation, these changes don't make sense. Is it possible that the dashed line in the last of the 3 sub-figures is instead a plot of PPD20V 1?

12) There are inconsistencies in the numbers reported. For example, the S.E. for the urban roadside TEOM vs Shinyei PPD20V is reported as 7.1 in Figure 3, and listed as 8 in Table 3, but the text (p.8, 2nd line) says that the Shinyei "sensor was within 4 $\mu$g/m3 of the TEOM". For the Atlanta rooftop, for the PPD60PV the S.E. is 7.1 in Figure 6, vs 17 in Table 3.

13) The added comparisons, in this revised version, between the Shinyei sensors and

a DustTrak in laboratory experiments seems inadequately examined. The advantage of using one source (incense) in repeated experiments is that it allows assessments of variations/inconsistencies between instruments. It also represents a best-case scenario of sorts for correlation strengths. However, a potentially substantial difference between comparing 1-min averages here, vs 1-hr averages in the field, is that lags in instrument response would much more greatly impact the 1-min comparisons. What is the characteristic response time for the Shinyei sensors?

Minor issues:

1) The abstract gives the impression that the study is much more comprehensive than it is. It should be edited to be more straightforward, e.g.

a. Replace "a number of select PM sensors" (line 11) with "three models of PM sensors"

b. Replace "a variety of ambient conditions and locations, including urban background..." (lines 11-12) with "a range of ambient conditions at 3 locations: urban background..."

c. Likewise, on p.2 (lines 33-34), "a variety of" should instead say "several", and "include several" should instead say "include 3 models of"

2) P.7, line 21 – the reference to "the way they were assembled in the junction box" is unclear. What was it about the assembly that might have led to a lack of correlation?

3) The figure captions need to be substantially expanded, with the location included for each plot, so that each one will stand on its own.

4) P.9, line 4 – the Williams citation is not included in the reference list.

5) Using a phrase like "most highly correlated" (p.12, line 3) seems inappropriate for an R2 value of 0.30. "Least poorly correlated", perhaps?

6) The abstract also gives an overly positive impression of the study's findings, with

the sentence (line 23) "The results of this work show the potential usefulness of these sensors for. . ."
* * *

---

## Referee Comment (RC3) · Anonymous Referee #1 · 10 Mar 2016

A much more technically rigorous assessment and thorough discussion of the limits of detection of the Shinyei low-cost PM sensors as pertains to the specific micro-environments to which they were deployed is necessary. Understanding sensor response to the bulk physio-chemical properties of the ambient PM distributions is fundamentally important and the rich 1-min data sets acquired by the authors hold some promise toward better informing the utility of these type of low-cost, IR, OPCs. The authors analyze and present a limited subset of a sparse, disparate experimental matrix. Laboratory-based, systematically-controlled calibrations for each of the low-cost PM sensors used here is a critically important pre-requisite to generating robust data handling protocols. Such laboratory assessments serve as the starting point for gen-

erating realistic error bars (exploiting best-case model PM scenario, minimum degrees of freedom in the experimental system). Such assessments must be completed prior to field deployment of low-cost sensors and subsequent interpretation of their data output. Through careful exploration of model PM distributions in the laboratory, the authors could begin to dis-entangle the complicating effects of particle size, refractive index, sensor-specific response (manufacturer or factory reproducibility (or lack thereof – changes in optical alignment), background levels (particle-free), and overall stability of sensor response over time in a constant PM concentration condition. The historical record of PM studies utilizing state-of-the-art characterization methods provides a template for what to expect in terms of the physio-chemical properties of near-roadside PM distributions, as well as developed and developing world urban background PM characteristics. Given this atmospheric intuition, the principle challenge in reconciling outputs from low-cost IR-based OPCs is the size detection limit of the device. If a given low-cost sensor can effectively measure 5% of the suspended 500 nm particles in a parcel of air, what mass fraction of the total PM2.5 is the device is able to detect? If the size distribution of the ambient PM is not static (i.e. dynamically changing from smaller to larger particles over the course of the day) what impact will that have on the PM2.5 mass fraction detected by the sensors? An underlying assumption in the correlation-approach utilized here is that the mass fraction of PM2.5 that the low-cost OPC is NOT detecting, remains constant. Based on the low R2 values reported in the manuscript and the microphysical processes governing PM emission and formation in the atmosphere, this missing mass fraction is most certainly not static. Interestingly, with larger size cut-offs for detection (1 um), the variability in the missing mass fraction may in fact decrease (especially in clean environments), improving correlations with co-located FRM and FEM. As written, the manuscript does not discuss their observations of low-cost OPC outputs in the context of the atmospheric PM2.5 distributions for each environment. The manuscript offers some important glimpses into the challenge of pollutant characterization with low-cost OPCs, but these insights do not comprise the bulk of the text or discussion.

I am in agreement with the comments of Anonymous Referees #3, #4, on all counts.

---

## Referee Comment (RC4) · Anonymous Referee #5 · 6 Apr 2016

**General comments**

The manuscript *Using Low Cost Sensors to Measure Ambient Particulate Matter Concentrations and On-Road Emissions Factors* promises to evaluate a number of low-cost PM sensors under a variety of conditions. However, I find several important problems with the methods employed in this work:

1) From my understanding of the text and photographs in Figures 1 and 2, PM measurements are performed by using a fan to blow ambient air over passive optical sensors. In my view, this is a very poor way to conduct particle sampling. What effect does the fan have on particle concentration and size distribution entering the sampling box? There is no way that the fan blades aren't acting as impactors and filtering particles

in some (unknown) way. The fans should have been on the exhaust end of the box pulling air through the sensors instead. Also, Figure 1 doesn't actually show where the inlet fan is located. Figure 2 seems to have circuit boards of different colors in the two photographs - this should be explained in the figure caption (are they different sensors or is this just an artifact of the photographs?).

2) The "calibration" presented here isn't really a calibration, but rather a correlation. Page 6 line 6 says that the entire dataset is used as a calibration - then what is used for analysis? You can't find the best regression between two datasets then plot the same data next to each other with the regression applied and say that they match well. In this case, they don't even match well anyways as many of the $R^2$ values are small.

3) Similar to #2, I have a problem with your basic assumption about what the sensor is measuring (page 32, line 19). You are equating the ratio of blocked laser time to total time as proportional to particle mass. This is not correct. These two may correlate with each other (and this paper shows that sometimes it does, but mostly its a poor correlation), but these values are not linked by any physics. The ratio you use is representative of total particle number concentration, not mass. To get mass, you need information about the size of the particles, which the sensors provide in a very primitive way, but you don't seem to be using this information. The use of this assumption may entirely explain why the correlations are so poor some of the time, but there is just not enough information in this paper to properly assess this.

4) The emission factor calculation would be a promising method if it were done more rigorously. It seems like only 1 short time period was hand-picked from the entire dataset because the data looked right and happened to give a number that fell between published values that span 2 orders of magnitude. As other reviewers have pointed out, the uncertainty on this calculation seems way too low and is, in fact, missing for the reference analyzers. There needed to be alot more supporting measurements (i.e. wind speed and direction) available as well to ensure this calculation is valid. To be truly beneficial to the community, as promised on Page 12 lines 10-13, this calculation

needs to be proven to be valid on much shorter averaging time periods and for many more test cases.

Having seen several other reviewer comments already posted, I am in agreement with these other reviewers on most points and will not repeat all of the same comments already presented. The authors should very carefully respond to each of their concerns as well.

**Specific comments**

While generally written well enough to be understandable, the manuscript does need some careful attention to detail in a few spots.

The abstract is written to sound very promising; however, many of the $R^2$ values are too low to be considered a positive result/correlation.

Several references are missing from the bibliography, including "EPA, 2015" and "Sensiron, 2010".

Page 2, line 10 - Can you really cite people's "desires"?

p 2, l 19 - What are the advantages and disadvantages? Be more specific.

p 2, l 33 - "variety" is actually just 3 different models from the same manufacturer; this is a bit misleading.

p 3, l 12 - How are the sensors promising?

p 3, l 29 - Did you talk to the manufacturers to try to get more information? To properly assess an instrument's performance, we really need to have more information on its design.

p 3, l 30 - Are the results supposed to be linear or exponential? On page 6, line 5 you

state that it doesn't matter whether Deming or simple linear regression is used - so what does this mean about the errors of each measurement? On page 8, line 2 you mention how a $5^{th}$ order polynomial has no physical meaning, but does an exponential fit have a physical meaning? Just because this is the shape of the signal near saturation does not mean that there is real meaning in that measurement range.

p 4, l 24 - "should have provided" - Did it? Be more specific.

p 5, l 7 - "Therefore" is basically saying that because these sensors can vary by a large amount because of varying particle composition in the ambient atmosphere, you are going to ignore controlled laboratory experiments and instead focus on field performance of these sensors. This would be an okay focus of the study *IF* you had more measurements to compare to and made proper assumptions (see #3 above). Otherwise, you are trying to evaluate sensors in an environment that they are expected to be highly varied (because particle composition is highly varied) and you are not measuring this varied composition with any other supporting measurements.

p 5, l 15 - A HEPA filter does not ensure that the TEOM is functioning properly. Be more specific and precise with the wording.

p 6, l 2 - Is $R^2 = 0.1$ really "marginal" correlation?

p 6, l 22 - If the entire sampling period is used to "calibrate" the PPD20V sensor to the TEOM measurements, it should not be surprising then that the absolute values of mass concentration are close.

p 6, l 23 - What does "tracked the TEOM well" mean, especially in light of how your 'calibration' was done?

p 7, l 22 - How did these effects likely lead to large errors? Be more specific.

p 8, l 12 - Why aren't the intercepts zero? Zero mass concentration should be zero voltage on the sensors, correct?
p 9, l 2-3 - I would say that the present study also shows that low-cost sensors do not perform well at US ambient concentrations.

p 11, l 9-10 - I do not understand this sentence.

Fig 5 - Is there a typo in the legends? The two PPD60PV curves look nothing like each other.

In general, more information could be given in each figure caption.

―――――――――――――――――――――

---

## Author Comment (AC1) · 27 May 2016

Please see attached responses to reviewer #3 followed by responses to other reviewers.

[Figure]

**Final Responses: Responses are bolded under each reviewer comment**

Referee #3:

General comments: The subject area of this study, low cost sensors for air quality measurements, is a very hot topic. It is therefore important papers in this area are written with great care to understand what new sensors can or cannot do, and a detailed analysis of the measurement data to check for cross-correlation and challenges. Unfortunately this paper has not realized this. The stated goal of the work was: "to evaluate a variety of lower cost alternatives for generating continuous pollutant measurements". However deploying the home-made box of sensors for between a few days up to a few weeks without replicates or complete data analysis of all the parameters is not particularly useful.

**We agree with the reviewer that this is an emerging research area of significant interest, and that careful evaluation of low cost sensors is important. While the field study testing period was constrained to a short period of time, we would argue that the unique testing environment – both urban United States and high concentration India environments – provide important evidence on sensor performance. These results will add to the growing body of work testing these and other sensors in a variety of environmental conditions.**

I would suggest the authors re-analysis all the data which they have recorded and undertake some studies, e.g. if you used 50% of the measurement period to calibrate the PM sensor, how well does it do against the other 50% of the dataset etc.

**We agree that this would be a useful evaluation, and was also mentioned by several of the reviewers. We have conducted additional analyses for the Hyderabad data using a few days of data to calibrate the data and then applying the calibration to the rest of the time period. The results are available in sections 3.1.3, Table 4, and Figure 6.**

Some further quality control experiments in the laboratory should also be done before re-submitting to peer review. The authors need to focus less on correlation plots and spend more time on the actual data, and the physical reasons for them, then more may be learned about how to do low cost measurements well.

**We agree that tests under controlled laboratory conditions provide some useful information on what drives the signal for low cost optical particle sensors, and we cite recent studies that have conducted that work (e.g., Austin et al., 2015; Wang et al., 2015). There are limitations in the ability to generate aerosol mixtures that match the variability of chemical and physical composition of particles in urban environments. This research study emphasizes the performance of sensors in real-world settings that represent areas that are likely to be of great interest for the deployment of sensors (e.g., urban areas near roads, high concentration areas in India). This work is meant to complement ongoing laboratory evaluations of optical particle sensors.**

Specific comments: Abstract The abstract almost wholly misrepresents the results of the study. Rather than reporting poor correlation of the sensors against the reference instrument, that area is almost completely ignored with a focus on emission factors. That part of the paper used less than two hours of data (with a correlation of 0.18 to a reference instrument) to conclude that emission factors could be measured with ~30% error. The conclusion that the paper's results has showed the potential usefulness

**Fig. 1.** Reviewer responses

**Performance of Low Cost Sensors Measuring Ambient Particulate Matter in High and Low Concentration Urban Environments**

Karoline K. Johnson[1], Michael H. Bergin[1], Armistead G. Russell[2], Gayle S. W. Hagler[3]

[1]School of Civil and Environmental Engineering, Duke University, Durham, NC, 27708, USA
[2] School of Civil and Environmental Engineering, Georgia Institute of Technology, Atlanta, GA, 30332, USA
[3] U.S. Environmental Protection Agency, Office of Research and Development, Research Triangle Park, NC, 27711, USA

*Correspondence to*: Karoline K. Johnson (Karoline.johnson@duke.edu)

**Abstract.** Air quality is a growing public concern in many countries, as is the public interest in having information on air pollutant concentrations within their communities. Quantifying the spatial and temporal variability of ambient fine particulate matter (PM$_{2.5}$) is of particular importance due to the potential health impacts associated with PM$_{2.5}$. This work evaluates three models of PM sensors (Shinyei: models PPD42NS, PPD20V, PPD60PV) in three locations: urban background (average PM$_{2.5}$: 8 µg m$^3$) and roadside sites in Atlanta, Georgia, USA (average PM$_{2.5}$: 21 µg m$^3$), as well as a location with substantially higher ambient concentrations in Hyderabad, India (average PM$_{2.5}$: 72 µg m$^3$). Additionally, a low cost carbon dioxide (CO$_2$) sensor (COZIR GC-0010) and a mid-cost black carbon sensor (microAeth AE51) were tested at the roadside in Atlanta. Low cost sensor measurements were compared against reference monitors at all locations. The PPD20V sensors had the highest correlation with the reference environmental beta attenuation monitor (E-BAM) with R$^2$ values above 0.80 at the India site while at the urban background site in Atlanta, the PPD60PV had the highest correlation with the tapered element oscillating microbalance (TEOM) with an R$^2$ value of 0.30. At the roadside site, only the PPD20V was used, with an R$^2$ value against the TEOM of 0.18. Although the results of this work show poor performance under lower USA concentrations, the results indicate the potential usefulness of these low cost sensors, including the PPD20V, for high concentration applications up to approximately 250 µg m$^3$. The CO$_2$ sensor had an R$^2$ value of 0.68 with the reference analyzer while the BC sensor correlated strongly to a multiangle absorption photometer (MAAP), with an R$^2$ of 0.99, at the Atlanta roadside site. These field testing results, although limited in nature, provide important insights into the varying performance of low cost particulate sensors used in highly contrasting atmospheric conditions and underlines the need to evaluate these emerging technologies, not only in the laboratory, but in their planned environment of application, prior to widespread use.

**1 Introduction**

Exposure to particulate matter (PM), particularly particles less than or equal to 2.5 micrometers in size (PM$_{2.5}$), is associated with a variety of adverse health impacts, including lung cancer (Laden et al., 2006), cardiovascular disease (Laden et al., 2006;Miller et al., 2007;Puett et al., 2009), and premature mortality (Puett et al., 2009). Although some cities in the US have PM values above the National Ambient Air Quality Standard (NAAQS) (EPA, 2013) annual PM$_{2.5}$ concentration value of 12

**Fig. 2.** Revised paper

[Figure]

**Fig. 3.** Revised paper with track changes

---

## Author Comment (AC2) · 27 May 2016

**Final Responses: Responses are bolded under each reviewer comment**

**Referee #3:**

General comments: The subject area of this study, low cost sensors for air quality measurements, is a very hot topic. It is therefore important papers in this area are written with great care to understand what new sensors can or cannot do, and a detailed analysis of the measurement data to check for cross-correlation and challenges. Unfortunately this paper has not realized this. The stated goal of the work was: "to evaluate a variety of lower cost alternatives for generating continuous pollutant measurements". However deploying the home-made box of sensors for between a few days up to a few weeks without replicates or complete data analysis of all the parameters is not particularly useful.

**We agree with the reviewer that this is an emerging research area of significant interest, and that careful evaluation of low cost sensors is important. While the field study testing period was constrained to a short period of time, we would argue that the unique testing environment – both urban United States and high concentration India environments – provide important evidence on sensor performance. These results will add to the growing body of work testing these and other sensors in a variety of environmental conditions.**

I would suggest the authors re-analysis all the data which they have recorded and undertake some studies, e.g. if you used 50% of the measurement period to calibrate the PM sensor, how well does it do against the other 50% of the dataset etc.

**We agree that this would be a useful evaluation, and was also mentioned by several of the reviewers. We have conducted additional analyses for the Hyderabad data using a few days of data to calibrate the data and then applying the calibration to the rest of the time period. The results are available in sections 3.1.3, Table 4, and Figure 6.**

Some further quality control experiments in the laboratory should also be done before re-submitting to peer review. The authors need to focus less on correlation plots and spend more time on the actual data, and the physical reasons for them, then more may be learned about how to do low cost measurements well.

**We agree that tests under controlled laboratory conditions provide some useful information on what drives the signal for low cost optical particle sensors, and we cite recent studies that have conducted that work (e.g., Austin et al., 2015; Wang et al., 2015). There are limitations in the ability to generate aerosol mixtures that match the variability of chemical and physical composition of particles in urban environments. This research study emphasizes the performance of sensors in real-world settings that represent areas that are likely to be of great interest for the deployment of sensors (e.g., urban areas near roads, high concentration areas in India). This work is meant to complement ongoing laboratory evaluations of optical particle sensors.**

Specific comments: Abstract The abstract almost wholly misrepresents the results of the study. Rather than reporting poor correlation of the sensors against the reference instrument, that area is almost completely ignored with a focus on emission factors. That part of the paper used less than two hours of data (with a correlation of 0.18 to a reference instrument) to conclude that emission factors could be measured with ~30% error. The conclusion that the paper's results has showed the potential usefulness

of low cost sensors for high concentrations is at best hopeful. The best sensor showed saturation at higher concentrations of PM. I would suggest the abstract be re-written to reflect the results presented.

**We have clarified the abstract to reflect the above comments. We are in agreement with the reviewer that the prior abstract did not adequately represent the results presented in the paper and have added wording regarding the overall performance of the sensors. Including page 1 line 19 "the results of this work show poor performance under lower USA concentrations". In addition we have removed the emissions factors work from this paper as it was a concern of many of the reviewers.**

P3 line 6 Section 2.1 There is a list of aims (i) –(iii). It is not clear having read the whole manuscript if (i) ((i) characterize three commercially available, relatively low-cost optical particle sensors) was done at all – the uncalibrated sensors appear to have been installed directly into "a sensor measurement package" (or box). Could the authors specific what (if anything) was done other than directly deploy the sensors and calibrate by running in parallel with other instruments?

**We have updated list for clarity, which is now written (Page 2 Line 30-32): "This research was conducted primarily through field studies designed to: (i) assess a sensor package capable of continuously measuring multiple air pollutants and (ii) characterize the performance of three commercially available, relatively low-cost optical particle sensors as well as a low cost $CO_2$ sensor and a mid-cost BC monitor compared to reference analyzers."**

Line 12 "The sensors appear promising after initial evaluation" Could this be referenced or detailed.

**We had conducted some very brief early testing of a variety of sensors in a laboratory and field environment, which led us to select the sensors shown in the paper for further field evaluation. Since the preliminary testing was brief and not included in the paper, this sentence will be removed for clarity.**

Line 13 and Table 1: The cost and manufacturer are not really the pertinent details required to evaluate a performance.

**Although not required for evaluating performance, we feel this information is important context for this discussion about low cost sensors.**

Table 1: Please could the authors add to the table summaries of the manufacturer specifications for the sensors.

**We have added this requested information to the updated Table 1.0.**

Line 13-14 and p6 line 1-7. Calibration: They state in line 30 a Deming regression is used to calibrate. Could they explain why they use this particular regression. The Deming regression assumes the errors for the two variables are independent (this is reasonable) and normally distributed (this is not known), and the ratio of their variances is known. (it is not known). The authors are not calibrating the sensors, they are using the reference instrument to calculate a PM from the sensor voltage. And they have assessed the performance by looking at the correlation of the sensor output against the reference TEOM for the same dataset. It would be more standard to calibrating with the TEOM and then comparing it against the TEOM for a different period – which is a reasonable experiment as long as calibration and measurement periods are clearly separated. IF they have done this, they have not communicated it in the manuscript. A flow diagram of their calibration and measurements would help.

**The reviewer brings up an important point about ratios of error varience assumptions used in the Deming regression approach that may make it not the best statistical method to apply. We have reexamined which statistical analysis may be most appropriate to apply. After careful consideration we have decided to first apply linear regression to calibrate the output from the sensor and then to apply orthogonal regression to minimize the errors in the X and Y directions. This methodology is detailed on pages 3-4 lines 32-3.**

**In addition, we have been careful in our use of the term "calibration" throughout the paper, and have also conducted the suggested analysis of the reviewer for the Hyderabad data by calibrating the data from the first few days and then applying the calibration to the rest of the data, which is provided in sections 3.1.3 (page 9, lines 2-20), Table 4, and Figure 6.**

P4 line 9: as per previous comment on Table 1, details of the T and RH sensor specifications should be included.

**We have updated the table as suggested by the reviewer (See Table 1).**

P4 line 17: the authors state that there is a flow of 67 l.m-1. An explanation of why such a high flow rate is used would be useful. What is the response time of the diffusive sampling sensor and then how does that related to the air mass sampled. It would be useful to see a flow diagram with theoretical response time (and empirical from the field measurements). I am slightly concerned that although the sensors are meant to diffusionally sample with the flow rates used, there would be a significant pressure differential on the inlet of the sensor volume and hence they would not be just diffusionally sampling.

**The sensors measure the light scattering from a volume (as compared to single particle scattering) and therefore sampling is not a function of flowrate (as compared to single particle sensing) as long as the flow is not negligible enough to generate diffusional and/or settling losses in the sampling volume. The flow is generally maintained in the sensors by a heated resistor that generates air flow based in a generated temperature and pressure differential. We have added some clarification in the text (page 3, lines 9-14).**

Is the internal box temperature equal to the external temperature in the field tests?

**We did not measure T outside of the boxes, although intend to do so in the future. We feel that it is the T and RH in the box that influences both the electronics and sensor performance which is why we chose to measure within the box.**

Line 20 In the multisensor unit the authors add 2 more fans: does this lead to a flow of 201 l.m-1? What effect might that have on the system performance? They note that the configuration means that the PPD42NS is in a different position and hence that may affect the results, but then do not mention it again even though PPD42NS is the worst performing of all the sensors AND (according to the introduction) most widely used. It seems little care was taken when trying to make sure an equivalence was ensured in the testing.

**The three fans were used so that each sensor had similar flow passing over them. This has been clarified in the text (page 5 lines 9-15): "The three fans provided ample flow through the PPD42NS and the $CO_2$ and temperature/RH sensors although not directly adjacent. Placing the PPD42NS further from the fan inlet allowed it to be further from the fan opening where stray light could enter and**

**influence the results. This is more important for the PPD42NS since it has a more open light scattering chamber than the other two sensors. With three fans, the exchange rate in the junction box was estimated to be approximately six times per second for the comparison box, although possibly less due to flow resistance through the box. Given that the sensors measure the light scattering from a volume it is not expected that the estimated PM concentrations are a function of flowrate, although it is possible that particles losses, particularly for larger coarse particles that can impact on surfaces within the sampling box, are influenced by the flow rates. We did not assess the dependence of air flow on particle losses, and an assumption is that fine particulate mass concentrations for the flow rates reported here are not influenced by particle losses."**

Pg 4 Line 27 this sentence does not make sense.

**We have clarified the statement as follows page 4 line z: " A 25 mm fan was positioned to draw air in to the instrument package and was positioned directly below the PM sensor. This was added to improve air flow through the sensor so that the sensors would be able to sample external air since the heating resistor would only supply flow through the individual PM sensors and not through the whole box."**

Line 29: "the addition of fans may change the size of particles drawn into the sensing volume": Did it? Were any trials of switching the fan on and off or box with and without lid done either in the lab or field?

**We can not be certain that the fans did not result in particle losses. Although the PM$_{2.5}$ calibrations were done with the fans inline as described in the text.**

P5 Section 2.2: This purpose of this paragraphs is difficult to understand. To paraphrase, the authors are saying: lab experiments have shown that different aerosol types give different results, so we assume they are not representative of field conditions so we decided to do a field experiment. Ambient aerosol is mixed! I am not sure whether the authors know what they are doing in this paragraph apart from imply that the lab experiments were not useful.

**This paragraph has been clarified so that the readers can understand that we have used field experiments because it is challenging to simulate ambient aerosol (Page 5, Line 17-26): "Particle properties are variable and are composed of both internal and external mixtures of chemical components that vary as a function of size. The response of optically-based PM sensors is largely a function of the actual properties of the ambient aerosol at the specific measurement location, including the size distribution and chemical composition. Lab studies with light scattering particle sensors have found the responses vary by a factor of 10-12 depending on particle size and composition (Wang et al. 2015, Austin et al. 2015). While laboratory evaluation is useful, there are limitations in the ability to generate aerosol mixtures that match the variability of chemical and physical composition of particles in urban environments. This work focuses mainly on field evaluations of sensors against reference monitors, rather than laboratory studies to evaluate sensor response as a function of particle size, composition, and concentration that is not representative of field conditions. However, we do discuss evaluations conducted in our laboratory as well as recent detailed laboratory analyses of similar sensors (Wang et al., 2015; Austin et al., 2015)."**

P 5 line 23: Was the TEOM inlet co-located with the sensor pack at 4 m. Did the TEOM have a PM10 or other size selecting head on it? If the inlets we not co-located what was the separation? There are strong gradients near roadsides therefore if you are testing one method against another these details should be noted. Similarly for the other two locations: Where is the TEOM or eBAM inlet c.f. the sensor package?

**Yes they were always within a few feet of each other. This has been clarified in the text.**

**Roadside: "The TEOM inlet was within a few feet of the sensor package" (Page 6, line 13-14).**

**Roof: The inlet of the TEOM and the sensors were located within about 3 meters of each other (page 6, lines 17-18).**

**Hyderabad: "The sensor package was attached to the E-BAM stand so they were measuring in the exact same location" (page 6, line 20).**

P6 line 9: Laboratory evaluation: The authors should either take this out or describe the experimental results fully. A table of the statistics from the experiment is not sufficient to describe the results. How did the three sensors co-vary with the DustTrak? A plot of the DustTrak concentration and the three sensors during the one hour experiment is the minimum needed for the reader to understand the outcomes. It does matter whether there is an offset, whether there is a response lag or if they all co-vary nicely.

**The reviewer request for more information regarding the laboratory test is understandable and we have expanded this section to show detailed results, which are now presented in Figure 9. The sensors respond similarly with some noise/scatter seen in the sensor data.**

Results section:

P6 line 28: inaccuracies are attributed to the TEOM when "using 1 hour versus 24 hour averages". This needs explaining t in more detail. The authors are not using 24 hour data.

**This comment was in reference to the EPA 24 hour National Ambient Air Quality Standard. We have removed this line to avoid confusion.**

The uncertainties in the TEOM are real and well characterized by many papers more recent than Allen et al., however stating they exist and looking at the dataset being assessed and working out the level of errors likely are two things. The particle masses being measured at the roadside are not very small – small is remote background sites where the mass is 0-5ug.m-3, not roadside. P7: lines 1- 11.

**I have added additional discussion of past research in this section including Carrico et al., 2003, Xu et al., 2004, Karagulian et al., 2012 and Kashuba and Scheff, 2008 (pages 7-8, lines 23-5).**

Because the authors have not included a plot which shows the variation of T, RH and the PM or CO2 in one figure it is almost impossible as a reviewer to comment on their statistics. They should have a figure with the data in it and then analyse it openly.

**The reviewer request for additional plots to more clearly see the multiple parameters simultaneously are understood. We had selected the original figures due to length concerns, however, have now added the requested additional figures (Figure 3, lines x-y)**

Again, how close were the two CO2 measurements? Were they exactly co-located?

**The $CO_2$ measurements were collocated with the PM sensors and inlets for the $CO_2$ measurments were within a few feet of each other. This has been updated in the manuscript (Page 8 line 5-6).**

Line 16: only the data from the PPD60V is shown in Figure 6. It would be useful to see all the data, despite the poor correlations.

**We agree with the reviewer, that showing all data may be useful. Figure 5 has been updated with scatter plots of the raw output from each sensor.**

Line 21. The authors imply in this sentence that if the RH>70% large errors are to be expected. Given the sensors are being proposed as ambient PM monitors and ambient RH is frequently above 70% RH, does this not imply the sensors are not fit for the purpose intended? The authors do not seem to have an accurate understanding of ambient atmospheric conditions and also do not spend any discussing this rather large potential problem.

**We agree that this is a misleading statement. There has been a great deal of work done on the RH dependence of light scattering and we have clarified the text with discussion and references. Generally speaking the enhancement of the light scattering coefficient by water uptake for anthropogenic aerosols ranges from around 10%-30% at an RH of 70% to 40-70% at an RH of 80% (Rood et al., 1987; McInnes et al., 1998). Although we have not specifically characterized the RH dependence of response of the sensors discussed in this paper, our comment was meant to point out the expected impact of RH on sensor response due to water uptake and light scattering enhancement. We have added clarifications in the text (page 4, lines 20-24).**

Line 30 onwards: The effect of sensor saturation is discussed only in terms of what curve to fit, rather than trying to understand why it is saturating and whether it is quantitative to use the data outside of the linear region. I suspect that you have multiple scatters and absorbers of light leading to a lower recorded concentration.

**We do not believe that there is a problem of multiple scattering in the viewing volume since these concentrations have been measured with convential nephelometery in many locations without such a response at similar high concentrations. We believe that the upper limit is related to the sensor electronic/optical configuration and could likely be avoided by changing the signal processing of the sensors, although we have not yet been able to determine if this is indeed the case.**

P8 and Figure 8: A high correlation coefficient is reported for the PPD20Vs at the Indian high PM concentration site. However inspecting the x-y plot, there is a lot of scatter in the <100ug.m-3 part of the graphs. Again it is a shame that the time series are not shown as only then can you really see the detail.

**Figure 6 has been added showing a time series of the sensor responses. In addition based on the comments of all the reviewers additional discussion of the performance of the sensors in Hyderabad using only low concentration data (<100 ug/m3) has been performed and is shown in Figure 8 is discussed in greater detail on page 9 lines 34-35.**

Unfortunately, the scatter plot does not tell a reader enough about the performance. The interesting information will be in the details which the authors have not shown or discussed. Again the authors

mention the 70% RH threshold as being important, however I would re-iterate that any sensor used in the environment must be able to function across the RH range.

**As we mentioned above there are certainly RH effects on sensor response. Below 70% RH they are likely less than 30%. Although future applications of such sensors to will need to determine RH response for a given region to accurately estimate PM above ~70%. In addition we have added additional time series of the sensors, their errors, and temperature and RH (Figure 6).**

P9 line 4: The statement that the sensors could perform better in future studies in an improved enclosure has no basis (though may possibly be correct). Why do the authors think this? What proposals for improvements are they proposing?

**It is possible the sensors could perform better in future studies in an improved enclosure with improved fan placement, better light interference protection (page 10, lines 30-31), and possibly other improved features. Multiple improvements have been suggested by multiple reveiwers.**

Line 17: Could the authors measure the temperature inside and outside of the box to address the question of the T variation of the electronics?

**We did not measure T outside of the boxes, although intend to do so in the future. We feel that it is the T and RH in the box that influences both the electronics and sensor performance which is why we chose to measure within the box.**

P10 line 4: Could the LOD as calculated by 2 or 3 SD of the noise on the blank signal be reported. I think this would be a better standard.

**We initially used this method to determine the LOD. But we feel this approach is not appropriate for the sensors discussed in this paper. This may give an unrealistic LOD.**

Also blanks as a function of temperature and RH would be interesting to see.

**We agree it would be interesting to see the zero response of the particle sensors with varying temperature and relative humidity, unfortunately we did not explicitly determine the T and RH dependence of the sensors for filtered air. As this study is emphasizing field evaluation, this detailed laboratory evaluation beyond the scope of this paper. It is worthwhile to point out that we did not see any obvious signs that the sensors drifted with these parameters.**

Is there any baseline drift on the sensors over the measurement periods?

**Other than the PPD42NS, there doesn't appear to be drift over these periods, although we can not say for certain. As mentioned above the changes during different periods seem to be mostly associated with variability in ambient concentrations as concentrations during the first few days of this study were much higher than during later during the study. This has been addressed further in the text (page 9, lines 14-20).**

Section 3.2: This is one of the most detailed sections of the paper, however in fact the authors only used just over 1 hours data out of 3 days deployment at the kerbside, ignored periods when $CO_2$ and PM did no correlate and came up with very good sounding numbers with high certainty (compared to all the other measurements in the paper). Unfortunately I do not feel the numbers are robust enough to publish, based on 1 hours worth of data with no replicates.

**Based on the comments from the 4 reveiwers we have decided to remove the emissions factors section of the paper and focus instead on the sensor performance.**

Section 4 Conclusions: The conclusions section seems to ignore all the poor statistical performances of the sensors, the potential environmental limitations, the issues with lack of correlation with reference instruments, the unknowns about sensor housing performance, the poor performances at < 100ug,m-3 and above 200 ug,m-3 and paints a very rosy summary. This section should be re-written to actually reflect the results reported.

**We have updated the conclusion to better reflect the results of the paper after removing the emissions factors section.**

**Referee #4**

This manuscript aims to assess the reliability of 3 low-cost Shinyei-brand PM sensors via comparisons with two other PM measurement methods, a TEOM and an E-BAM, for 3 urban locations. While not clearly explained in the mspt, the output from the Shinyei sensors apparently is uncalibrated ("raw"), so sensor reliability was assessed based on how strong the correlation was between the Shinyei's raw output and the reference monitor's reported mass concentration. For a small set of lab experiments, the Shinyei sensors were compared with a DustTrak. The authors also utilized a low-cost CO2 sensor (COZIR GC-0010), showing a comparison with a reference monitor for one location. While a "mid-cost" MicroAeth BC sensor was also deployed, no comparisons to reference monitors were made in this study.

**Since deleting the emissions factors section of this paper we have added an additional plot summarizing the performance of the Microaeth as compared to the MAAP (Figure 3.C, page 8, lines 12-16).**

Characterizing the performance of new monitoring devices under field conditions is a worthwhile goal. However, the number of sites measured (3), the small range of concentrations seen at 2 of the sites, and the analytical insights seem insufficient to contribute to this goal. The only insight this reader gained from the manuscript is that the Shinyei sensors are unreliable—the correlation with reference monitors is much too low to view the devices as even semi-quantitative. In addition, given that they fail to capture temporal trends some of the time (p.6, line 25), they cannot be viewed as qualitatively reliable, either. I am not sure if this modest insight (which was already known for one of the models), by itself, merits publication. If it does, then there are a number of other major issues needing to be addressed, as itemized below.

**While the field study testing period was constrained to a short period of time, we would argue that the unique testing environment – both urban United States and high concentration India environments – provide important evidence on sensor performance. These results will add to the growing body of work testing these and other sensors in a variety of environmental conditions. We hope that we have addressed all the reveiwers comments to their satisfaction**

Major issues: 1) The manufacturer reports different lower particle size detection limits for different models (ranging from 0.5 – 1 μm; page 3, line 28). For a motor vehicle-dominated site, one would expect much of the mass to be submicron – this would lead to differences in measurements for the different models, as well as with the TEOM (which uses a filter with a very high capture efficiency for submicron

particles). The mspt needs to discuss how/why a sensor that only detects down to 1 μm would be effective at estimating EF values for combustion emissions.

**It is not clear how the manufacturers determined the particle sizing lower limits for the different sensors, and we were not able to get more specific information form them on this point. The sensors use the principle of volume scattering and the sensor signal should therefore scale roughly with the aerosol light scattering coefficient. The light scattering coefficient depends on many factors (particle size, refractive index, and wavelength of the light), and for urban Atlanta we found a clear link between light scattering coefficient and PM2.5 ($R^2$=0.8) (Carrico et al., 2003) with roughly 60% of the light scattering by particles greater than 0.5 um. A tremendous advantage of the using a volume scattering approach is that generally speaking the scattering coefficient is highly correlated with the accumulation mode mass with smaller particles (less than 0.1 um) and larger particles (greater than 2-3 um) generally do not dominate light scattering due to their much lower mass scattering efficiencies. So to summarize, as far as we can tell the mentioned particle size ranges are somewhat arbirtrarlily defined by the manufacturer. We did generate incense smoke in our chamber studies as described below that is dominated by particles in the 0.1um – 0.5um size range (by mass) and the sensors clearly responded.**

2) While I did not see this mentioned until p.8 (line 33), it appears that the Shinyei sensors were measuring total PM, with no size cut. I presume that the E-BAM was operated with a PM2.5 inlet, although this is never stated in the mspt. However, I believe the model 1400a TEOM is designed for PM10 measurements. The authors need to explain why the plots, and much of the results/discussion, list PM2.5 as what is being measured or estimated.

**Both reference analyzers used a $PM_{2.5}$ inlet and we have used the sensors to estimate $PM_{2.5}$ (although they are not size selective). This has been updated in the manuscript. "Both reference analyzers were operated with a $PM_{2.5}$ inlet cyclone. Although the sensors are not size selective we have compared them against a $PM_{2.5}$ reference since providing a surrogate measurement for $PM_{2.5}$ is envisioned to be the common application for these low cost sensors." (pg 3 line 16-18)**

3) The discussion of biases in the TEOM measurements seems inadequate (pp. 6-7). There have been quite a few papers since Allen et al's 1997 paper comparing the C2 TEOM with other sensors, including those relying on light scattering measurements. As just one example, Karagulian et al (JEM 14:2145, 2012) found an R2 value of 0.75 between the TEOM and a SidePak. It would seem important to note how well previous comparisons between TEOMs and other light scattering instruments have worked, as context for (and comparison with) the measurements from the Shinyei sensors.

**I have added additional discussion of past research in this section including Carrico et al., 2003, Xu et al., 2004, Karagulian et al., 2012 and Kashuba and Scheff, 2008 (pages 7-8, lines 23-5).**

4) How does the level of air pre-heating for the Shinyei sensors compare with that for the TEOM? While heating in the TEOM is mentioned as a possible artifact on p.6 (line 30), there is no mention at this point in the mspt that the Shinyei sensors are also heated (even though this was stated back on p.3 line 14).

**The reviewer makes a good point here and this statement has been removed. We measure the temperature in the boxes and assume that this temperature is representative of that within the sensor light scattering volumes.**

5) How did the timing of disagreements between PM monitors (p. 6, line 24) relate to the RH measurements (or the temperature measurements)?

**At the roadside both temperature and humidity had no relation as shown in the figures below:**

[Figure]

6) For the Hyderabad measurements, clearly the R2 value is driven by the highest concentration data points. If only values 0 µg/m3 are used, does the R2 become similar to the Atlanta rooftop?

**Although we are unsure of exactly the concentration range this reviewer was looking for, we have updated the paper with additional discussion of the performance of the sensors in Hyderabad using only low concentration data (<100 ug/m3). This is shown in Figure 8 is discussed in greater detail on page 9 lines 34-35.**

7) The entire "Estimating emission factors" section is inadequately supported, and should be omitted, due to the following concerns: a. The R2 for this site (between PPD20PV and the TEOM) was 0.18, which is a very uncertain starting point for relying on the PPD20PV values in calculations.

b. As is mentioned in this revised version of the mspt, the light scattering characteristics of the PM will vary between "background urban" and vehicle emissions. So applying a single regression to the Atlanta roadside, where only a subset of the data is believe to be vehicle-dominated, will lead to even more inaccuracies.

c. The EF estimate is based on a 5-min period of elevated PM and CO2 data, even though 1-min measurements were collected over 3 days. Oddly, looking at the CO2 data (Figure 4), it appears that the 5-min period chosen is in the midst of a ~6 hr period where CO2 measurements were ~480-550ppm. But the synchronous (Figure 3) show PM levels fluctuating much more frequently over this same time period.

d. No evidence is provided that the approach to choosing this 5-min period involved objective (=statistical) analyses – the authors only say (p.11, line 17) that they chose a morning rush hour (even though there would have been 3 morning rush hours of data), "where both the pollutant and CO2 rose and fell at the same time". Why weren't more time periods tested?

e. The lack of information on wind direction, traffic density, and "background" concentration levels (upwind of the roadway) leads to more uncertainties and unsupported assumptions.

f. After obtaining an EF value, and deducing that this would correspond to about 30% of the fuel combustion involving diesel, on a freeway that should have been "dominated by gasoline-fueled vehicles" (p.11, line 4), the mspt says (p.11, line 22) that this EF value "is likely high". g. Then, a 2nd EF is found, based on BC and involving a different time period (8-9pm), with no justification for why this period was expected to have high concentrations of roadway emissions. This EF corresponds to 13% diesel combustion – but there is no discussion of whether this EF value is trustworthy or believable. Is it feasible that there could be this large a portion of diesel vehicles making local deliveries, in the evening?

h. I am extremely skeptical that the confidence bounds shown with the EF values are accurate – they seem much too low, and the method used to quantify uncertainty isn't clearly explained. By SE, are you giving the standard error of estimate, representing the 85% (that is, 1 sigma) prediction interval around each best fit line? But how are you accounting for the fact that, for the roadside site, only 18% of the variability in the Shinyei measurements can be accounted for by the reference PM measurements?

**After careful considerations of all options we have decided to leave the emissions factors part of this paper out.**

8) On p.14, it is noted that the optics weren't maintained at all during the "few week" deployment in India. How do the authors know that the sensors were still performing acceptably near the end of the deployment period? No temporal measurements are shown or were discussed for the India deployment, nor were tests performed to assess how frequently optics maintenance was needed.

**Only the performance of the PPD42NS appears to decrease over the time periods shown. The other sensors do not appear to show degredation. We have added additional time series in Figure 6.**

9) The use of an exponential (monotonically increasing) eqn to capture a saturation type effect seems misleading – it gives the erroneous impression that, despite saturation, an accurate mass concentration can still be inferred. If saturation was indeed a problem, then all measurements greater than a certain raw output level should have been omitted from the fitting protocol, and subsequent analyses.

**We do not believe that there is a problem of multiple scattering in the viewing volume since these concentrations have been measured with convential nephelometery in many locations without such a response at similar high concentrations. We believe that the upper limit is related to the sensor electronic/optical configuration and could likely be avoided by changing the signal processing of the sensors, although we have not yet been able to determine if this is indeed the case.**

10) The exact same set of data used to determine the best fit line (or exponential) was then transformed, using this best fit equation. It does not seem scientifically appropriate to apply a transformation equation to the exact same set of data that was used to find the equation.

**We agree that this could be misleading and was also mentioned by several of the reviewers. We have conducted additional analyses for the Hyderabad data using a few days of data to calibrate the data and then applying the calibration to the rest of the time period. The results are available in sections 3.1.3, Table 4, and Figure 6.**

11) In Figure 5, there are noticeable discrepancies between the "raw output" data for the PPD60PV and the µg/m3. In some cases, e.g., around the 12/11 and 12/13 tick marks, large variations in the raw data appear to have been almost completely smoothed out in the µg/m3 plot. In other cases, e.g., around the 12/15 tickmark, a large spike has appeared in the µg/m3 plot that is completely absent in the raw output plot. Given the monotonic nature of the fitting equation, these changes don't make sense. Is it possible that the dashed line in the last of the 3 sub-figures is instead a plot of PPD20V 1?

**This point of confusion is due to a figure labeling error on our part. We have corrected the confusing plot, which should have been labeled the PPD42NS graph. Thanks to the reviewer for catching this typo.**

12) There are inconsistencies in the numbers reported. For example, the S.E. for the urban roadside TEOM vs Shinyei PPD20V is reported as 7.1 in Figure 3, and listed as 8 in Table 3, but the text (p.8, 2nd line) says that the Shinyei "sensor was within 4 µg/m3 of the TEOM". For the Atlanta rooftop, for the PPD60PV the S.E. is 7.1 in Figure 6, vs 17 in Table 3. 13)

**We have clarified the statistics throughout this paper trying to focus on errors (Reference-sensor) and the 2 standard deviations of those errors (95% confidence).**

The added comparisons, in this revised version, between the Shinyei sensors and a DustTrak in laboratory experiments seems inadequately examined. The advantage of using one source (incense) in repeated experiments is that it allows assessments of variations/inconsistencies between instruments. It also represents a best-case scenario of sorts for correlation strengths. However, a potentially substantial difference between comparing 1-min averages here, vs 1-hr averages in the field, is that lags in instrument response would much more greatly impact the 1-min comparisons. What is the characteristic response time for the Shinyei sensors?

**Although we did not measure the characteristic response time for the Shinyei, we have added a plot of the timeseries during the chamber testing (Figure 9). We did not notice a significant lag between the response of the sensors and the response of the dustrak as illustrated in the figure.**

Minor issues: 1) The abstract gives the impression that the study is much more comprehensive than it is. It should be edited to be more straightforward, e.g. a. Replace "a number of select PM sensors" (line 11) with "three models of PM sensors" b. Replace "a variety of ambient conditions and locations, including urban background. . ." (lines 11-12) with "a range of ambient conditions at 3 locations: urban background. . ."

**This is a good point and has been changed (Page 1, Lines 10-11) "This work evaluates three models of PM sensors (Shinyei: models PPD42NS, PPD20V, PPD60PV) under a range of ambient conditions in three locations:"**

c. Likewise, on p.2 (lines 33-34), "a variety of" should instead say "several", and "include several" should instead say "include 3 models of"

**This is also a good point and has been updated (page 2 line 25).**

2) P.7, line 21 – the reference to "the way they were assembled in the junction box" is unclear. What was it about the assembly that might have led to a lack of correlation?

**This was refering to fan and sensor placement is explained in greater detail elsewhere in the paper so this line has been removed for clarity.**

3) The figure captions need to be substantially expanded, with the location included for each plot, so that each one will stand on its own.

**All figure captions have been updated to better represent their contents.**

4) P.9, line 4 – the Williams citation is not included in the reference list.

**This has now been added.**

5) Using a phrase like "most highly correlated" (p.12, line 3) seems inappropriate for an R2 value of 0.30. "Least poorly correlated", perhaps?

**Changed to: "most correlated"**

6) The abstract also gives an overly positive impression of the study's findings, with the sentence (line 23) "The results of this work show the potential usefulness of these sensors for…"

**The abstract has been updated to better reflect our results. We have added wording regarding the overall performance of the sensors, in agreement with the reviewer that the prior abstract did not adequately represent the results presented in the paper. In addition we have removed the emissions factors from this paper as it was a concern of many of the reviewers.**

**Referee #1**

A much more technically rigorous assessment and thorough discussion of the limits of detection of the Shinyei low-cost PM sensors as pertains to the specific microenvironments to which they were deployed is necessary. Understanding sensor response to the bulk physio-chemical properties of the ambient PM distributions is fundamentally important and the rich 1-min data sets acquired by the authors hold some promise toward better informing the utility of these type of low-cost, IR, OPCs. The authors analyze and present a limited subset of a sparse, disparate experimental matrix.

**A primary objective of the paper was to determine the feasibility of using such sensors for urban locations within the US, with a focus being those influenced by roadway emissions, and a developing region of the world having relatively high particulate concentrations. They represent relatively low and high concentrations over which such sensors will be used.**

Laboratory-based, systematically-controlled calibrations for each of the low-cost PM sensors used here is a critically important pre-requisite to generating robust data handling protocols. Such laboratory assessments serve as the starting point for generating realistic error bars (exploiting best-case model PM scenario, minimum degrees of freedom in the experimental system). Such assessments must be completed prior to field deployment of low-cost sensors and subsequent interpretation of their data output. Through careful exploration of model PM distributions in the laboratory, the authors could begin to dis-entangle the complicating effects of particle size, refractive index, sensor-specific response (manufacturer or factory reproducibility (or lack thereof – changes in optical alignment), background levels (particle-free), and overall stability of sensor response over time in a constant PM concentration condition.

**These field results will add to the growing body of work testing these and other sensors in a variety of environmental conditions. Some lab evaluations have been done by other groups (Austin et al. 2015, Wang et al. 2015). At this time we do not feel these particular sensors perform well enough to warrant extensive lab testing.**

The historical record of PM studies utilizing state-of-the-art characterization methods provides a template for what to expect in terms of the physio-chemical properties of near-roadside PM distributions, as well as developed and developing world urban background PM characteristics. Given this atmospheric intuition, the principle challenge in reconciling outputs from low-cost IR-based OPCs is the size detection limit of the device. If a given low-cost sensor can effectively measure 5% of the suspended 500 nm particles in a parcel of air, what mass fraction of the total PM2.5 is the device is able to detect? If the size distribution of the ambient PM is not static (i.e. dynamically changing from smaller to larger particles over the course of the day) what impact will that have on the PM2.5 mass fraction detected by the sensors?

**The light scattering coefficient depends on many factors (particle size, refractive index, and wavelength of the light), and for urban Atlanta we found a clear link between light scattering coefficient and PM2.5 ($R^2$=0.8) (Carrico et al., 2003) with roughly 60% of the light scattering by particles greater than 0.5 um. A tremendous advantage of the using a volume scattering approach is that generally speaking the scattering coefficient is highly correlated with the accumulation mode mass with smaller particles (less than 0.1 um) and larger particles (greater than 2-3 um) generally not dominating light scattering due to their much lower mass scattering efficiencies.**

An underlying assumption in the correlation-approach utilized here is that the mass fraction of PM2.5 that the low-cost OPC is NOT detecting, remains constant.

**The primary assumption is that the mass scattering efficiency, $E_{scat}$ (defined as the light scattering coefficient divided by the PM2.5 mass concentration) remains constant. This also implies that most of the light scattering is from particles with diameters less than 2.5 um. Our past work has shown that the mass scattering efficiency tpypically only varies by ~30% in urban Atlanta (Carrico et al., 2003). Also in a rural region of China with local coal and biomass burning mass scattering efficiencies were also similar to those reported for urban Atlanta (Xu et al., 2004). So it appears that anthropogenic emissions relatively near to sources generally result in aerosols particles with similar mass scattering efficiencies. Although we can not say for certain why this is, it is likely due to the fact that emitted particles and related precursors relatively quickly grow into accumulation mode aerosols, with smaller particles (less than 0.1 um) contributing little to mass and scattering. As noted in Xu et al. (2004), the mass scattering efficiency does change when dust dominates particle mass, but this was not the case for our measurements. We have added discussion in the text.**

Based on the low R2 values reported in the manuscript and the microphysical processes governing PM emission and formation in the atmosphere, this missing mass fraction is most certainly not static. Interestingly, with larger size cut-offs for detection (1 um), the variability in the missing mass fraction may in fact decrease (especially in clean environments), improving correlations with co-located FRM and FEM.

**For the reasons mentioned above we do not believe that having a 1 um cut size (or 2.5 um cut size as was the case for the TEOM and EBAM) would change the correlations.**

As written, the manuscript does not discuss their observations of low-cost OPC outputs in the context of the atmospheric PM2.5 distributions for each environment. The manuscript offers some important glimpses into the challenge of pollutant characterization with low-cost OPCs, but these insights do not comprise the bulk of the text or discussion. I am in agreement with the comments of Anonymous Referees #3, #4, on all counts

**The reviewer raises a number of insightful points regarding the challenges of characterizing and interpreting the signals produced by a low cost sensor. We agree with the reviewer on many points – that laboratory characterization under controlled conditions is highly valuable and that interpretation of the sensor performance needs to consider the differing physicochemical composition of aerosols in the two environments. We see this proposed paper as presenting evidence regarding real-world performance of these low-cost sensors, and cite existing laboratory studies that provide complementary analysis. As there are a number of significant entities that are jumping to quickly deploying these types of sensors en masse, we feel these early field tests provide some important insight as to whether the sensors provide any reasonable comparison to true fine particulate mass measurements. We appreciate the comments by all of the reviewers and also would ask this reviewer to assess our comments to the other reviewers to insure that he/she is in agreement with our replies.**

**Referee #5**

The manuscript Using Low Cost Sensors to Measure Ambient Particulate Matter Concentrations and On-Road Emissions Factors promises to evaluate a number of low-cost PM sensors under a variety of conditions. However, I find several important problems with the methods employed in this work:

1) From my understanding of the text and photographs in Figures 1 and 2, PM measurements are performed by using a fan to blow ambient air over passive optical sensors. In my view, this is a very poor way to conduct particle sampling. What effect does the fan have on particle concentration and size distribution entering the sampling box? There is no way that the fan blades aren't acting as impactors and filtering particles in some (unknown) way. The fans should have been on the exhaust end of the box pulling air through the sensors instead. Also, Figure 1 doesn't actually show where the inlet fan is located. Figure 2 seems to have circuit boards of different colors in the two photographs - this should be explained in the figure caption (are they different sensors or is this just an artifact of the photographs?).

   **This is the exact same equipment just photographed in 2 ways. I have included an arrow to the fan. Fan placement is something we will improve in future work. Although we can not say exactly what impact the fans had on the sampling design we did calibrate them with the fans in place. In future tests we will move the fans to the exhaust position.**

2) The "calibration" presented here isn't really a calibration, but rather a correlation. Page 6 line 6 says that the entire dataset is used as a calibration - then what is used for analysis? You can't find the best regression between two datasets then plot the same data next to each other with the regression applied and say that they match well. In this case, they don't even match well anyways as many of the R2 values are small.

**"Calibration" has been clarified throughout and we have added calibration of the first half of the data applied to the second half for the Hyderabad data which is provided in sections 3.1.3 (page 9, lines 2-20), Table 4, and Figure 6.**

3) Similar to #2, I have a problem with your basic assumption about what the sensor is measuring (page 32, line 19). You are equating the ratio of blocked laser time to total time as proportional to particle mass. This is not correct. These two may correlate with each other (and this paper shows that sometimes it does, but mostly its a poor correlation), but these values are not linked by any physics. The ratio you use is representative of total particle number concentration, not mass. To get mass, you need information about the size of the particles, which the sensors provide in a very primitive way, but you don't seem to be using this information. The use of this assumption may entirely explain why the correlations are so poor some of the time, but there is just not enough information in this paper to properly assess this.

**This sensor is measuring light scattering not blocked laser time. The primary assumption in using these sensors is that the mass scattering efficiency, $E_{scat}$ (defined as the light scattering coefficient divided by the PM2.5 mass concentration) remains constant. This also implies that most of the light scattering is from particles with diameters less than 2.5 um. Our past work has shown that the mass scattering efficiency typically only varies by ~30% in urban Atlanta (Carrico et al., 2003).**

4) The emission factor calculation would be a promising method if it were done more rigorously. It seems like only 1 short time period was hand-picked from the entire dataset because the data looked right and happened to give a number that fell between published values that span 2 orders of magnitude. As other reviewers have pointed out, the uncertainty on this calculation seems way too low and is, in fact, missing for the reference analyzers. There needed to be a lot more supporting measurements (i.e. wind speed and direction) available as well to ensure this calculation is valid. To be truly beneficial to the community, as promised on Page 12 lines 10-13, this calculation needs to be proven to be valid on much shorter averaging time periods and for many more test cases. Having seen several other reviewer comments already posted, I am in agreement with these other reviewers on most points and will not repeat all of the same comments already presented. The authors should very carefully respond to each of their concerns as well.

**After careful consideration we have decided to remove the emissions factors discussion from this paper.**

Specific comments

While generally written well enough to be understandable, the manuscript does need some careful attention to detail in a few spots. The abstract is written to sound very promising; however, many of the R2 values are too low to be considered a positive result/correlation.

**The abstract has been modified to reflect the suggestions of this reviewer and all other reviewers.**

Several references are missing from the bibliography, including "EPA, 2015" and "Sensiron, 2010".

**The citations have been double checked and now all are included.**

Page 2, line 10 - Can you really cite people's "desires"?

**Good point. Should have included the citation with the previous sentence : (page 2, lines 10-11) "but the high costs associated with conventional measurements limit the number of air quality monitoring sites globally, leading to generally sparse spatially-defined air quality information that may not represent actual exposures (Stevens et al., 2014). Citizens and policy makers desire more data to make decisions for individual and societal health and well-being."**

p 2, l 19 - What are the advantages and disadvantages? Be more specific.

**This has been removed along with the emissions factors part of this paper.**

p 2, l 33 - "variety" is actually just 3 different models from the same manufacturer; this is a bit misleading.

**We have revised the text to clarify that three sensors were tested (page 2, lines 24-29)**

p 3, l 12 - How are the sensors promising?

**We have removed the word promising throughout the paper.**

p 3, l 29 - Did you talk to the manufacturers to try to get more information? To properly assess an instrument's performance, we really need to have more information on its design.

**We reached out the manufacturers to gain as much information as possible regarding the sensor design. The paper contains all of the knowledge we have gained – manufacturers have generally been reluctant to share information they consider proprietary in nature.**

p 3, l 30 - Are the results supposed to be linear or exponential?

**We would expect them to be linear though other studies have seen non-linear responses. This has been clarified in the text. (page 3-4, lines 32-5).**

On page 6, line 5 you state that it doesn't matter whether Deming or simple linear regression is used - so what does this mean about the errors of each measurement?

**This was a poor statistical explaination. Correlation of the data is not related to the linear function that is applied to it. Based on the comments of other reviews we have updated our calibration method. We have decided to first apply linear regression to calibrate the output from the sensor and then to apply orthogonal regression to minimize the errors in the X and Y directions. This methodology is detailed on pages 3-4 lines 32-3.**

On page 8, line 2 you mention how a 5th order polynomial has no physical meaning, but does an exponential fit have a physical meaning? Just because this is the shape of the signal near saturation does not mean that there is real meaning in that measurement range.

**This is a good point. We have removed the comment about physical meaning.**

p 4, l 24 - "should have provided" - Did it? Be more specific.

**We have removed this line for clarity.**

p 5, l 7 - "Therefore" is basically saying that because these sensors can vary by a large amount because of varying particle composition in the ambient atmosphere, you are going to ignore controlled laboratory experiments and instead focus on field performance of these sensors. This would be an okay focus of the study IF you had more measurements to compare to and made proper assumptions (see #3 above). Otherwise, you are trying to evaluate sensors in an environment that they are expected to be highly varied (because particle composition is highly varied) and you are not measuring this varied composition with any other supporting measurements.

**We have improved this paragraph based on the comments of all reveiwers. The goal of this paper is to evaluate low cost sensors in a way that they could be easily deployed in multiple environments. This would not be accomplished if we were using other instruments to gather additional information.**

p 5, l 15 - A HEPA filter does not ensure that the TEOM is functioning properly. Be more specific and precise with the wording.

**We agree with this comment and have updated this line as follows (page 5-6, line 30-2): "A high efficiency particle arresting (HEPA) filter was attached at the inlet on the TEOM periodically to ensure that there were no leaks in the sampling line. Data and any instrument error flags were reviewed periodically and that the instrument was functioning properly."**

p 6, l 2 - Is $R^2$ = 0.1 really "marginal" correlation?

**The word marginal has been removed since it could be misleading.**

p 6, l 22 - If the entire sampling period is used to "calibrate" the PPD20V sensor to the TEOM measurements, it should not be surprising then that the absolute values of mass concentration are close.

**We agree that this was a misleading statement and have updated it as follows (page 7, lines 14-15): "The PPD20V was within 8.3 μg m$^{-3}$ of the TEOM at an hourly average 95 % of the time ($s_d$)."**

p 6, l 23 - What does "tracked the TEOM well" mean, especially in light of how your 'calibration' was done?

**This was a misleading phrase and has been removed.**

p 7, l 22 - How did these effects likely lead to large errors? Be more specific.

**We have added additional discussion about the effects of RH throughout the paper (page 4, lines 20-24). "Based on past work characterizing the change in light scattering coefficient as a function of RH for anthropogenic aerosol it is expected that water uptake on aerosol particles will result in an increase in the light scattering cofficient of 10%-30% at an RH of 70% RH, and 40%-70% at an RH of 80%(Rood, 1987;McInnes et al., 1998). Therefore higher RH measurements may result in overestimates of PM mass by the sensors since the reference measurements are for dried aerosol and do not reflect substantial amounts of water mass."**

p 8, l 12 - Why aren't the intercepts zero? Zero mass concentration should be zero voltage on the sensors, correct?

**Although this is true for the digital sensor, it is possible that an analog signal may produce a non-zero signal when there is zero concentration due to electronic noise.**

p 9, l 2-3 - I would say that the present study also shows that low-cost sensors do not perform well at US ambient concentrations.

**We would agree with this statement and have tried to update the manuscript throughout to reflect this. Overall this was a misleading line and has been expanded. "Previous studies using low cost ($150-$2050) scattering PM sensors at US ambient concentrations (~0-30 µg m$^{-3}$) have had max R$^2$ with FEMs of 0.8 and min R$^2$ of 0. This paper also looked into temperature and RH artifacts (Williams, 2014)" (page 10, lines 27-29).**

p 11, l 9-10 - I do not understand this sentence.

**This has been removed along with the emissions factors section of the paper.**

Fig 5 - Is there a typo in the legends? The two PPD60PV curves look nothing like each other. In general, more information could be given in each figure caption.

**The close review is appreciated and the confusion was due to a labeling error, which we have corrected. Additional explanation has been added to the figure captions.**

---

## Author Comment (AC3) · 27 May 2016

Please see attached manuscript.

Please also note the supplement to this comment:
http://www.atmos-meas-tech-discuss.net/amt-2015-331/amt-2015-331-AC3-supplement.zip